# Lipopolysaccharide stimulates dynamic changes in B cell metabolism to promote proliferation

Dana MS Cheung[1], Momchil Razsolkov[1], Fabrizia Bonacina[2], Stephen Andrews[1], Megan C Sumoreeah[1], Linda V Sinclair[1], Andrew JM Howden[1], J Simon C Arthur[1]*

[1]Division of Cell Signalling and Immunology, Faculty of Life Sciences, University of Dundee, Dundee, United Kingdom; [2]Department of Pharmacological and Biomolecular Sciences "Rodolfo Paoletti", University of Milan, Milan, Italy

## eLife Assessment

Findings from this study are considered **fundamental** because they identify amino acid uptake, cholesterol synthesis, and protein prenylation as key metabolic regulators of B cell activation, proliferation, and survival, advancing understanding of T-independent immune responses. The study links metabolic reprogramming directly to B cell function, highlighting how cellular metabolism supports immune fitness. The evidence is **compelling**, combining unbiased proteomic profiling with genetic and pharmacological validation to demonstrate causal roles for these pathways.

*For correspondence:
j.s.c.arthur@dundee.ac.uk

Competing interest: The authors declare that no competing interests exist.

## Abstract

Naive B cells exit quiescence and enter a proliferative state upon activation, ultimately differentiating into antibody-secreting or memory B cells. Toll-like receptor (TLR) ligands, such as lipopolysaccharide (LPS), can serve as physiological stimuli to initiate this transition. Using quantitative proteomics, we show that TLR4 engagement induces metabolic reprogramming in murine B cells, increasing the expression of amino acid transporters and cholesterol biosynthetic enzymes. The amino acid transporter SLC7A5 is markedly upregulated following LPS stimulation, and conditional deletion of *Slc7a5* impairs B cell proliferation, underscoring its essential role in B cell activation. LPS also elevates intracellular cholesterol levels, and inhibition of the rate-limiting enzyme HMG-CoA reductase blocks proliferation. This effect was mediated by a dual requirement for cholesterol metabolism and protein prenylation downstream of HMG-CoA reductase. Notably, this was not unique to TLR4 signalling but is also observed in B cells activated via TLR7, TLR9, CD40, or the B cell receptor. Together, these findings reveal that metabolic rewiring, including amino acid uptake and cholesterol metabolism, is an essential feature of B cell activation and proliferation.

## Introduction

B cell activation can be initiated through either T-cell-dependent (TD) or T-cell-independent (TI) pathways, with TI activation further subdivided into a type I or type II response depending on the stimuli. During TD activation, following recognition of a protein antigen via the B cell receptor (BCR), the B cell can present peptides from the antigen to CD4 T cells, resulting in T cell activation. Additionally, the B cell receives co-stimulation through the binding of its CD40 receptor to the CD40 ligand found on T cells, which promotes differentiation, proliferation, somatic hypermutation, and class switch recombination (CSR) (*Hoffman et al., 2016*). Type I TI responses are activated by agonists such as lipopolysaccharide (LPS), which bind to toll-like receptors (TLRs) on B cells to induce a strong proliferative signal and increase antibody production. Longer-term exposure to LPS can also promote the differentiation

of plasma cells (*Genestier et al., 2007*). Conversely, type II TI antigens consist of long repetitive structures, such as polysaccharides, which at high concentrations crosslink multiple BCRs to stimulate proliferation and antibody production (*Obukhanych and Nussenzweig, 2006*).

A well-established model to study TI B cell activation is to use LPS, a component of the cell wall of gram-negative bacteria, which stimulates TLR4 (*Hoshino et al., 1999*; *Poltorak et al., 1998*) to induce robust B cell proliferation and antibody secretion, both ex vivo (*Coutinho et al., 1974*; *Dziarski, 1982*) and in vivo (*Johnson et al., 1956*). Upon activation, TLR4 propagates downstream signalling through the adaptor proteins MyD88 and TRIF. This in turn activates pathways, including MAPK and NF-κB signalling, to elicit the required physiological response (*Arthur and Ley, 2013*; *Kawai et al., 2024*). The essential role of TLR4 in LPS-induced activation of B cells has been demonstrated by studies using TLR4-deficient mice. B cells from TLR4 or MyD88 knockout mice did not proliferate in response to LPS or lipid A stimulation ex vivo (*Hoshino et al., 1999*; *Kawai et al., 1999*). Similarly, in studies where B cells from MyD88- and TLR4-knockout mice were transferred into µMT mice (that lack mature B cells), a significant reduction in IgG1 production was observed (*Pasare and Medzhitov, 2005*).

Naive B cells have been shown to be in a quiescent state with low metabolic demand (*Waters et al., 2018*). Conversely, B cell activation must require a significant increase in biomolecules such as proteins, lipids, and nucleic acids to support their rapid expansion and proliferation. The existing literature has focused on the role of glucose and glucose transporters in B cell function. Previous studies have demonstrated that activation using LPS or anti-IgM to stimulate the BCR increases glucose uptake and leads to an increased oxygen consumption rate (*Dufort et al., 2007*; *Jellusova et al., 2017*). Whether this corresponds to changes in metabolic enzymes at a proteomic level, and whether TI B cell activation utilises other metabolic pathways, remains unknown. This increased glucose uptake has been shown to provide precursors for ribonucleotide and lipid synthesis (*Waters et al., 2018*). Similarly, the deletion of *Slc2a1* (which encodes for the glucose transporter GLUT1) impaired germinal centre formation, plasma cell differentiation, and the production of high-affinity antibodies (*Bierling et al., 2024*; *Brookens et al., 2024*). Recent reviews have expressed the need for studies investigating metabolic changes in activated B cells, especially the role of lipid metabolism, which is poorly understood (*Johnstone et al., 2024*; *Peeters and Jellusova, 2024*).

T cell activation has been shown to stimulate global changes in metabolism at a proteomic level (*Procaccini et al., 2016*; *Howden et al., 2019*; *Tan et al., 2017*). However, there are no comparable proteomic studies that demonstrate how B cells support the change in metabolic requirements and energy demand from quiescence to activation upon LPS stimulation. Therefore, we used Data Independent Acquisition (DIA)-based mass spectrometry to profile the B cell response to a combination of LPS + Interleukin 4 (IL-4) stimulation, a well-established ex vivo model of TI B cell activation. From this, we identified a substantial upregulation of proteins involved in multiple metabolic pathways, including amino acid uptake and cholesterol biosynthesis. We demonstrate the requirement of L-amino acid transporter SLC7A5 for B cell survival and proliferation ex vivo. Similarly, we highlight the necessity of both cholesterol metabolism and protein prenylation downstream of the rate-limiting enzyme HMG-CoA reductase for B cell growth, survival, and proliferation. These results highlight the pronounced metabolic rewiring induced downstream of LPS + IL-4 stimulation.

## Results

### LPS + IL-4 stimulation of murine B cells promotes proliferation and changes the expression of enzymes involved in cellular metabolism

Previous studies have shown that ex vivo stimulation of murine B cells with LPS and IL-4 can drive B cell proliferation and class switching to IgG1 (*Snapper et al., 1988*). Consistent with this, we found that lymphocytes isolated from lymph nodes and subsequently stimulated with LPS + IL-4 resulted in the proliferation of CD19 +B cells, as judged by dilution of Cell Trace Violet (CTV) labelling. LPS + IL-4 stimulation led to extensive proliferation, with up to six generations of B cells present after 72 hr (*Figure 1A*). Proliferation itself is thought to be a prerequisite for cells to undergo class-switching (*Greenwood et al., 2006*). In line with this, only LPS + IL-4 stimulated B cells found in later generations were positive for IgG1 (*Figure 1A*). On a similar note, cell size is considered a key regulator of the G1/S transition, as progression through the cell cycle can only continue once a certain threshold in size has been reached (*Barberis et al., 2007*). Accordingly, we found that 72 hr of LPS + IL-4

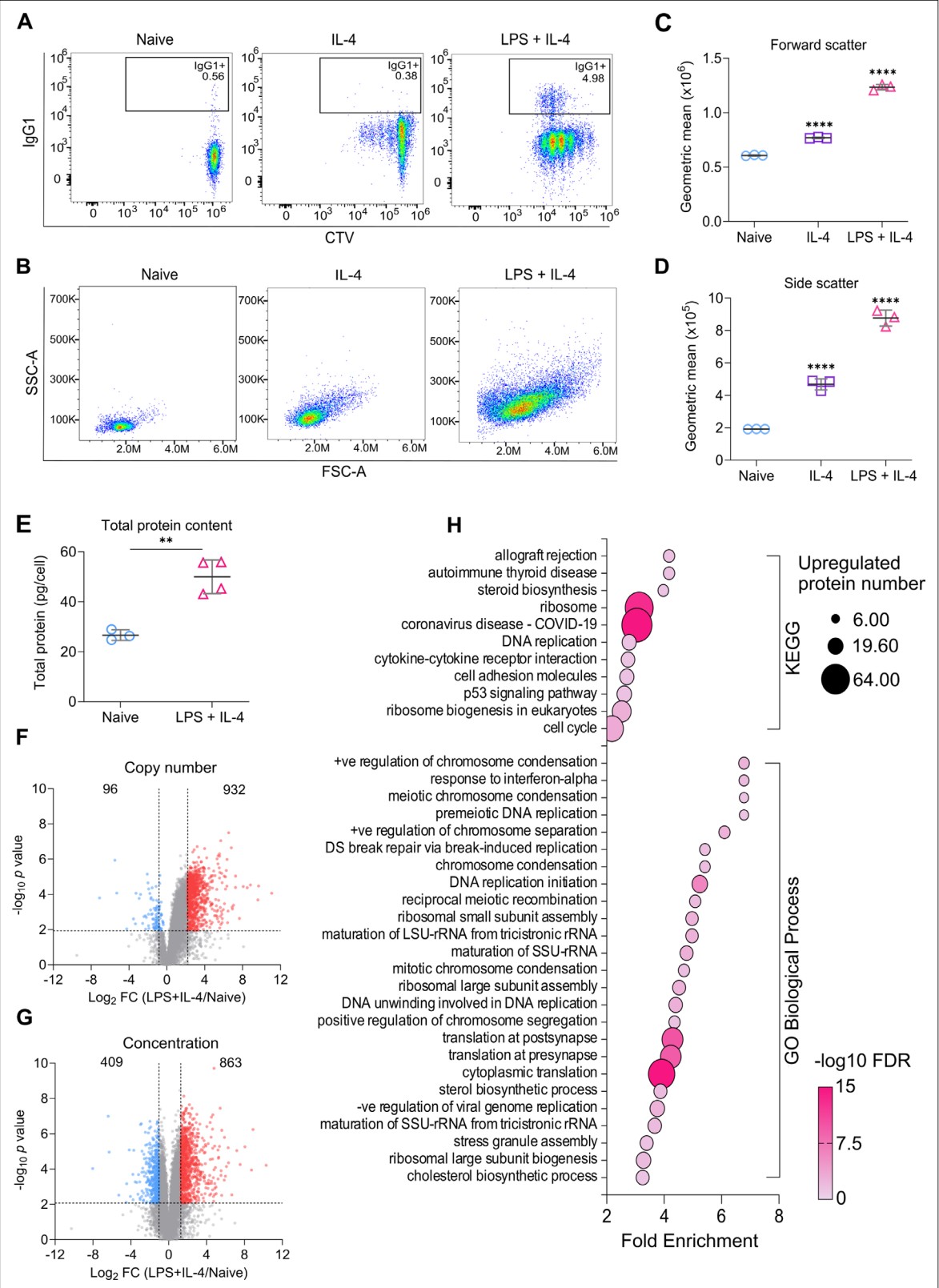

**Figure 1.** Stimulation with LPS + IL-4 promotes B cell proliferation and class switch recombination. (**A–D**) Cells from the lymph nodes of C57BL6/J mice were stained with Cell Trace Violet (CTV) and stimulated with IL-4 (10 ng/ml) +/- LPS (20 µg/ml) for 72 hr and analysed by flow cytometry. Data for naive B cells (stained on day 0) is also shown. Live CD19+B cells were identified using the gating strategy described in *Figure 1—figure supplement 4A*. (**A**) Representative flow cytometry plots comparing IgG1 expression and CTV staining after 72 hr of IL-4 or LPS + IL-4 stimulation. (**B**) Representative plots

*Figure 1 continued on next page*

*Figure 1 continued*

for FSC and SSC at 72 hr of IL-4 or LPS + IL-4 stimulation with geometric means for the (**C**) forward scatter and (**D**) side scatter. Data shows the results of three biological replicates. Data was analysed by one-way ANOVA followed by multiple comparison testing via Sidak's analysis. For comparisons to naive B cells, $p<0.0001$ is indicated by ****. Full ANOVA results are given in **Supplementary file 5**. (**E–H**) Cells from the lymph nodes of C57BL6/J mice were stimulated with LPS (20 µg/ml) and IL-4 (10 ng/ml) for 24 hr, then CD19+B cells were isolated by FACS. Alternatively, naive CD19+B cells were sorted directly from ex vivo lymph node cells. Cells were lysed and analysed by proteomics as described in the methods. Samples from four mice for LPS + IL-4 and three mice for naive were generated. (**E**) Total protein content (pg/cell) was estimated from the proteomic data. Statistical power was determined using an unpaired two-tailed Student's t-test, where $p<0.01$ is indicated by **. (**F**) Volcano plot depicting changes in estimated protein copy number in naive vs LPS + IL-4 stimulated B cells. (**G**) Volcano plot showing the estimated cellular protein concentration (µM) in naive vs LPS + IL-4 stimulated B cells. Horizontal dashed lines indicate $q<0.05$. Vertical dashed lines indicate $\log_2$ fold change of one standard deviation away from the median. (**H**) Enrichment analysis of the upregulated proteins in (**G**) against GO-term and KEGG databases.

The online version of this article includes the following source data and figure supplement(s) for figure 1:

Source data 1. Raw FACS files for *Figure 1A-D* (72 hr LPS + IL-4 stimulation lymph nodes).

Figure supplement 1. Unstimulated B cells do not proliferate or undergo class switching.

Figure supplement 1—source data 1. Raw FACS files for *Figure 1—figure supplement 1A–B* (24 vs 72 hr unstimulated lymph nodes).

Figure supplement 2. LPS + IL-4 stimulation upregulates glycolysis in B cells.

Figure supplement 2—source data 1. Raw FACS files for *Figure 1—figure supplement 2G–H* (MitoTracker staining).

Figure supplement 3. LPS + IL-4 stimulation does not increase STAT1 phosphorylation.

Figure supplement 3—source data 1. Raw FACS files for *Figure 1—figure supplement 3B–C* (p-STAT1 staining).

Figure supplement 4. Representative gating strategies.

Figure supplement 4—source data 1. Raw FACS files for *Figure 1—figure supplement 4A* (lymph node gating strategy), 4B (MitoTracker gating strategy), 4 C (cell cycle analysis gating strategy), 4D (OPP gating strategy).

stimulation led to a marked increase in cell size and granularity as indicated by a higher forward/side scatter compared to naive B cells (*Figure 1B–D*). Stimulation with IL-4 alone resulted in a smaller increase in cell size than a combination of LPS + L-4, but did not give rise to a strong proliferative response or class switching (*Figure 1A–D*). Naive B cells were used for comparison in ongoing experiments, as surviving unstimulated B cells exhibited a similar phenotype in the flow cytometry analysis; however, the lack of stimulation led to significant cell death from 24 hr onwards (*Figure 1—figure supplement 1*).

These functional differences following activation suggest a significant change may be occurring at the protein level. To investigate this further, we analysed changes in the proteome after 24 hr of stimulation, as this time point was sufficient for B cells to increase in size but retain their synchrony in terms of the number of divisions they had undergone ex vivo. DIA-based mass spectrometry was used to compare the proteomes of naive B cells to activated B cells after 24 hr of LPS + IL-4 stimulation. The proteomic ruler method (*Wiśniewski et al., 2014*) was then used to estimate protein copy numbers and concentration. Using this approach, a total of 8030 protein species were identified, of which 7187 were quantified in both naive and LPS + IL-4 stimulated B cells. Based on the mass spectrometry data, the predicted total protein mass in the B cells increased by approximately two-fold following stimulation with a combination of LPS + IL-4 (*Figure 1E*). This change in protein mass was due to an increased copy number of most proteins in the cell (*Figure 1F*), with a median $\log_2$ fold increase of 1.04. To adjust for the increase in cell size, the fold change in protein concentration was analysed, as this provides an indication of what proteins are selectively upregulated or downregulated as opposed to those that scale with cell size.

For the data using estimated cellular concentrations, using cutoffs of $q<0.05$ and a log2 fold change more than one standard deviation away from the median, 863 proteins were found to be upregulated following activation, while a further 301 were detected in all of the stimulated B cells but none of the naive ones. 409 proteins were downregulated in the LPS + IL-4 stimulated B cells in comparison to naive B cells, while a further 57 were detected in the naive B cells, but none of the activated B cells (*Figure 1G*).

To identify the biological pathways that were affected by LPS + IL-4 stimulation, proteins that were classified as regulated using the criteria above were searched against Gene Ontology (GO) Biological Processes terms and *Kyoto Encyclopaedia of Genes and Genomes* (KEGG) databases. Enrichment analysis on the downregulated proteins did not highlight any specific processes. However, upregulated

proteins were associated with terms related to the cell cycle, ribosome biogenesis, protein translation, and steroid biosynthesis (*Figure 1H*, *Supplementary file 1*).

While changes in the above processes would make sense in terms of cell proliferation, it was notable that changes in the glycolytic and oxidative phosphorylation pathways were not highlighted in this analysis. Previous studies have indicated that there is an increased demand for glucose in LPS-activated B cells (*Dufort et al., 2007*; *Johnstone et al., 2024*). Isotope tracing experiments following CD40L and IL-4 activation to stimulate proliferation have suggested that this increased glucose uptake is used in the pentose phosphate pathway, potentially to provide NADPH and precursors for nucleotide biosynthesis, while alternative carbon sources may be used to drive an increase in the TCA cycle and oxidative phosphorylation (*Waters et al., 2018*). The levels of enzymes involved in glucose metabolism were therefore examined in the proteomic data. In general, changes in the estimated concentrations of proteins involved in glycolysis, pentose phosphate pathway, TCA cycle, or oxidative phosphorylation were not observed (*Figure 1—figure supplement 2A,B*), although their copy numbers did increase in proportion to the increase in cell size (*Figure 1—figure supplement 2C,D*). B cells express the SLC2A1 (Glut1) glucose transporter, and deletion of this gene has been shown to reduce B cell activation in vivo (*Bierling et al., 2024*; *Brookens et al., 2024*). There was an increase in the concentration of SLC2A1, which would be consistent with an elevated ability to take up glucose (*Figure 1—figure supplement 2E*). In addition, there was a 13-fold increase in hexokinase 2 (HK2), one of the enzymes involved in catalysing the 1st committed step for glucose to enter the glycolytic or pentose phosphate pathways (*Tanner et al., 2018*) (*Figure 1—figure supplement 2F*) and has recently been shown to be required for maximal B cell responses to ex vivo stimulation with LPS (*Paradoski et al., 2024*). To investigate if the increased copy number of Electron Transport Chain proteins correlated with changes in mitochondrial volume, B cells were stained with MitoTracker Red, a fluorescent dye that binds to mitochondria. We found that MitoTracker staining was significantly increased following 24 hr of LPS + IL-4 stimulation (*Figure 1—figure supplement 2G,H*).

Notably, one of the enriched processes was identified as the 'response to interferon alpha'. To take this further, we compared our data with a previously published RNAseq dataset that looked at the effect of IFNα on murine B cells (*Mostafavi et al., 2016*). We found that the majority of the IFNα regulated genes were not significantly upregulated at a protein level following LPS + IL-4 stimulation compared to naive B cells (*Figure 1—figure supplement 3A*). IFNα and IFNβ signalling requires phosphorylation of the transcription factor STAT1 to induce transcription of type I IFN-dependent genes (*Platanias, 2005*). We found that LPS stimulation did not lead to detectable phosphorylation of STAT1 compared to stimulation with IFNβ (*Figure 1—figure supplement 3B–C*). Together, this suggests that there is not a strong interferon response in this system. It is possible, however, that as we stimulated lymph node cells before sorting for B cells for the proteomic experiment, a low level of type I interferon production from myeloid cells may have occurred, which could have contributed to a weak IFN gene signature in the B cells.

## LPS + IL-4 stimulation upregulates proteins involved in the cell cycle

Stimulation with LPS + IL-4 causes B cells to exit from a quiescent state and start to proliferate (*Figure 1*). This was confirmed using DAPI staining, which demonstrated that while naive B cells were predominantly in a G0/1 state, after 24 hr of LPS + IL-4 stimulation, the proportion of B cells in the S and G2 stages of the cell cycle were increased (*Figure 2A and B*). Progression through G1 is associated with an increase in Cyclin D expression, and activation of CDK4 and 6 (*Rubin et al., 2020*). At the same time, inhibitory cell cycle proteins such as p27Kip1 (Cdkn1b) are downregulated (*Wagner et al., 1998*). Therefore, we analysed the proteomic data for evidence of these changes. In naive B cells, Cyclin D was not detected, while levels of CDK4 and 6 were low (*Figure 2C–E*). The levels of Cyclin D, CDK4, and CDK6 were all increased following stimulation (*Figure 2C–E*). Naive B cells also expressed high levels of p27Kip1, and this was reduced by LPS + IL-4 stimulation (*Figure 2F*).

During the cell cycle, unphosphorylated RB1 binds to the E2F1/2/3 transcription factor, preventing the transcription of genes required for cell cycle progression. Increased CDK4/6 activity initiates RB1 phosphorylation, reducing its interaction with E2F, and resulting in the induction of E2F target genes, allowing progression through G1 to the S phase of the cell cycle (*Engeland, 2022*; *Rubin et al., 2020*). Increased phosphorylation of RB1 was observed in B cells following LPS + IL-4 stimulation (*Figure 2G and H*). In line with the increased RB1 phosphorylation, the levels of protein expressed from known

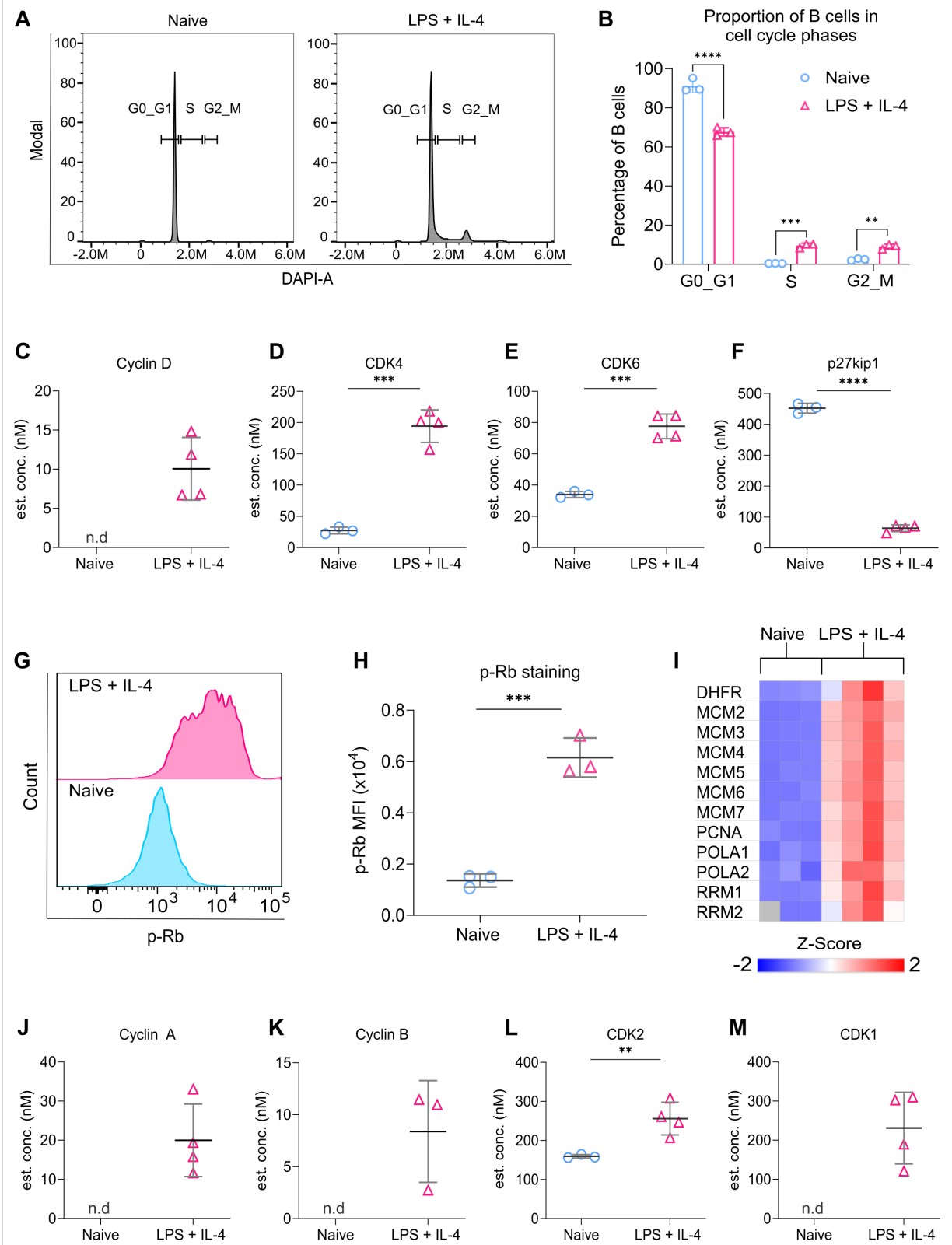

**Figure 2.** Proteins involved in cell cycle progression are upregulated in LPS + IL-4 activated B cells. (**A–B**) B cells were purified from the spleens of C57BL6/J mice and either fixed on isolation (naive) or stimulated with LPS (20 µg/ml) and IL-4 (10 ng/ml) for 24 hr before fixation. Cells were then stained with DAPI and CD19. The cell cycle stages were analysed using the gating strategy shown in *Figure 1—figure supplement 4C*. (**A**) Representative histograms showing the proportion of B cells in different phases of the cell cycle. (**B**) Quantification shows three technical replicates from cells isolated

*Figure 2 continued on next page*

*Figure 2 continued*

from one mouse and is representative of three independent experiments. Statistical power was determined using two-way ANOVA followed by multiple comparison testing via Sidak's analysis, where $p < 0.01$ is indicated by **, $p < 0.001$ by *** and $p < 0.0001$ by **** for comparisons between the naive and LPS + IL-4 conditions. (**C–F**) Graphs depicting changes in the estimated cellular concentration (nM) of proteins implicated in entry into the cell cycle, determined from the proteomic dataset described in *Figure 1*. (**C**) Cyclin D, (**D**) CDK4, (**E**) CDK6, (**F**) p27Kip1. (**G–H**) B cells were purified from the spleens of C57BL6/J mice and stimulated with LPS (20 µg/ml) and IL-4 (10 ng/ml) for 24 hr before fixing and staining for phospho-retinoblastoma (**p–Rb**). Gating strategy for p-Rb staining is shown in *Figure 1—figure supplement 4C*. (**G**) Representative histogram comparing p-Rb staining in naive and LPS + IL-4 stimulated B cells. (**H**) Quantification shows three technical replicates from cells isolated from one mouse and is representative of three independent experiments. (**I**) Heat map showing the expression of proteins encoded by E2F target genes, derived from the proteomic data. (**J–M**) Graphs depicting changes in the estimated cellular concentration (nM) of proteins implicated in cell cycle progression, determined from the proteomic dataset described in *Figure 1*. (**J**) Cyclin A, (**K**) Cyclin B, (**L**) CDK2, (**M**) CDK1. P values were determined using an unpaired two-tailed Student's t-test for (**H**) or represent adjusted p values from the FDR calculations applied to the proteomic dataset (**D, E, F, L**), where $p(adj) < 0.01$ is indicated by **, $p(adj) < 0.001$ by ***, and $p(adj) < 0.0001$ by ****.

The online version of this article includes the following source data for figure 2:

**Source data 1.** Raw FACS files for *Figure 2A–B* (DAPI staining) and *Figure 2G–H* (p-Rb staining).

E2F target genes, such as *Mcm2-7, Pcna, Dhfr, Rrm1/2,* and *Pola*3 (*Chicas et al., 2010*; *Engeland, 2022*; *Leone et al., 1998*; *Wagner et al., 1998*) were increased following LPS + IL-4 stimulation (*Figure 2I*). Further progression through the cell cycle requires Cyclins E, A, and B, as well as CDK2 and CDK1. While Cyclin E was not detected in the proteomic data, the levels of Cyclin A, Cyclin B, CDK2, and CDK1 were all increased by LPS + IL-4 stimulation (*Figure 2J–M*).

## Protein synthesis in LPS + IL-4 stimulated B cells is dependent on the uptake of amino acids

The cell growth and proliferation promoted by LPS + IL-4 stimulation would be expected to require an increased rate of protein synthesis. This is consistent with the proteomic analysis, which demonstrated an increase in total protein content within activated B cells (*Figure 1E*) and an enrichment of proteins associated with protein translation, including proteins that regulate ribosome biogenesis and the ribosomes themselves (*Figure 1H*). The ribosome is composed of the 40 S and 60 S subunits, each of which is composed of multiple proteins (*Doudna and Rath, 2002*). LPS + IL-4 stimulation increased the total concentration of ribosomal proteins that make up both the 40 S and 60 S subunits (*Figure 3A and B*). This overall increase was due to higher expression of most of the individual proteins that make up the subunits, rather than a large increase in one or two of the proteins (*Figure 3C and D*). In line with the increase in ribosomal proteins, there was also an upregulation of proteins associated with ribosome biogenesis (*Figure 3E–G*). To determine if this correlated with an increased rate of protein synthesis, we measured the incorporation of the puromycin analogue o-propargyl-puromycin (OPP) into newly synthesised proteins. Consistent with the upregulation of ribosomal proteins, OPP incorporation was significantly higher in B cells stimulated with LPS + IL-4 for 24 hr compared to naive B cells (*Figure 3H–I*). Since LPS + IL-4 stimulation increased OPP incorporation, we tested whether this was a stimulus-specific effect or whether other stimuli known to activate B cell proliferation could increase protein synthesis. Murine B cells also respond to agonists of TLR7 (*Weeratna et al., 2005*) and TLR9 (*Arunkumar et al., 2013*; *Yi et al., 1998*). Both the TLR7 agonist Resiquimod and the TLR9 agonist CpG increased OPP incorporation in B cells (*Figure 3—figure supplement 1*). Naive B cells can also be activated ex vivo by ligation of their BCR with anti-IgM antibodies or by stimulation with CD40 ligand (CD40L). Similar to TLR agonists, both stimuli also increased OPP incorporation in B cells (*Figure 3—figure supplement 1*).

Elevated protein synthesis during rapid proliferation would require an increased supply of amino acids. B cells have also been reported to increase amino acid biosynthesis upon stimulation. scRNAseq has shown that cycling B cells express higher levels of mRNA for enzymes in the serine biosynthetic pathway (*D'Avola et al., 2022*). Furthermore, the level of these proteins, as well as rates of serine biosynthesis, were found to be increased by stimuli such as anti-IgM, CD40L, and CpG, which are known to induce B cell proliferation (*D'Avola et al., 2022*). While this was not further addressed in the current study, the proteomic data reported here did show that serine biosynthetic enzymes, including the rate-limiting enzyme PHDGH, were increased in response to LPS + IL-4 stimulation (*Figure 4—figure supplement 1*).

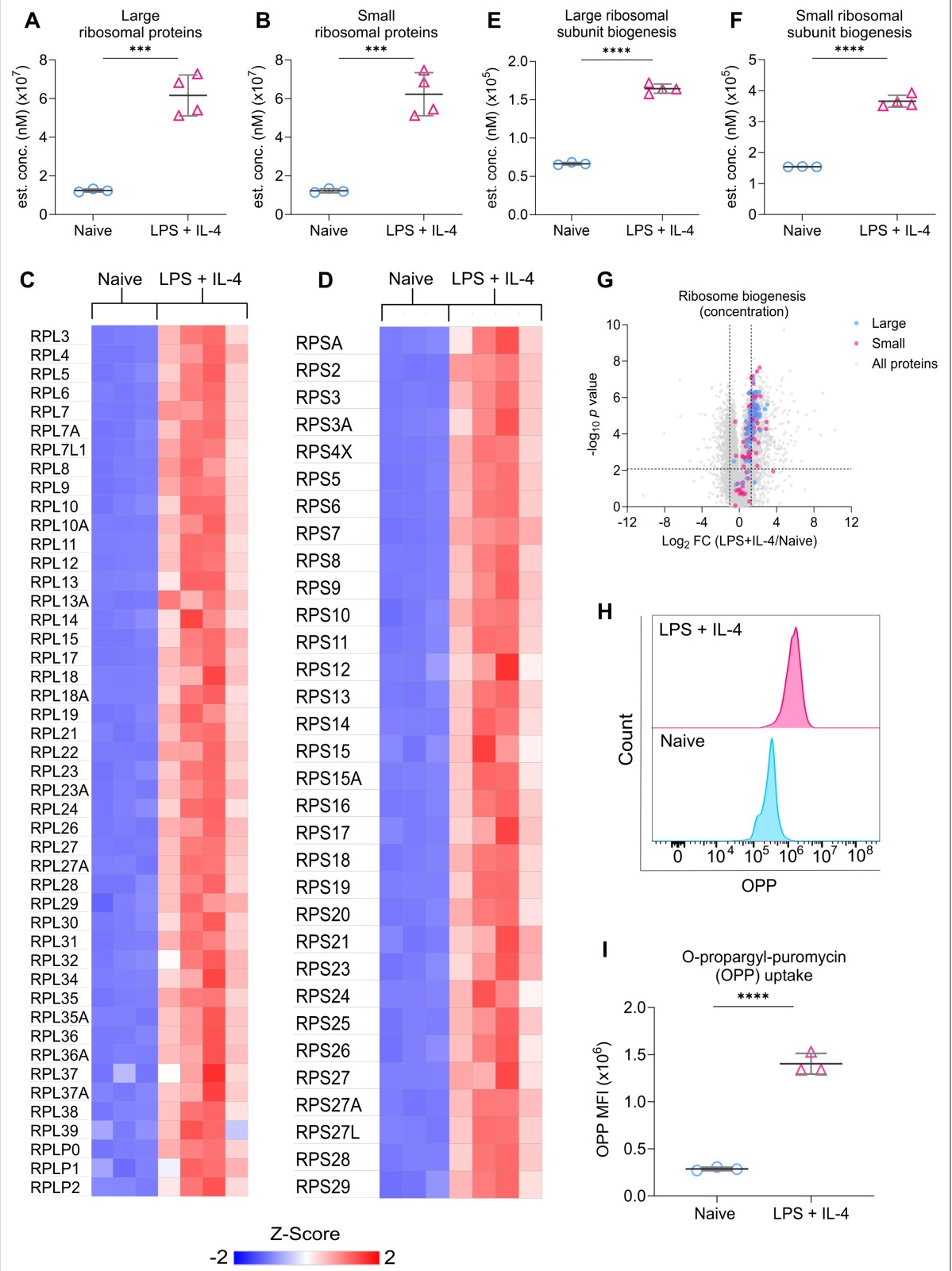

**Figure 3.** LPS + IL-4 stimulation promotes protein synthesis. (**A, B**) Graphs show changes in cellular concentration (nM) of the sum of proteins that make up the large (**A**) and small (**B**) ribosomal subunits, with heat maps showing the expression of the individual proteins making up the (**C**) large and (**D**) small subunits. Adjusted p values from the FDR calculations applied to the proteomic dataset, where *p(adj)*<0.001 is indicated by ***. (**E–G**) The proteomic dataset was mined for proteins involved in the biogenesis of the large and small ribosomal subunits based on the GO terms: GO:0000027, GO:0042273,

*Figure 3 continued on next page*

*Figure 3 continued*

GO:0000028, GO:0042274. (**E**) shows the sum of the proteins involved in the biogenesis of the large subunit and (**F**) shows the small subunit, with individual proteins represented on the volcano plot (**G**). Horizontal dashed lines on (**G**) indicate $q<0.05$ while vertical dashed lines indicate $\log_2$ fold change more than one standard deviation away from the median. Adjusted p values from the FDR calculations applied to the proteomic dataset, where $p(adj)<0.0001$ is indicated by ****. (**H–I**) Splenocytes from C57BL6/J mice were stimulated with LPS (20 µg/ml) and IL-4 (10 ng/ml) for 24 hr before fixing and staining for the uptake of puromycin analogue O-propargyl-puromycin (OPP) to measure protein synthesis. Gating strategy for OPP staining is shown in *Figure 1—figure supplement 4D*. (**H**) Representative histogram comparing OPP uptake between naive and LPS + IL-4 stimulated B cells. (**I**) Quantification shows three technical replicates from cells isolated from one mouse. Statistical power was determined using an unpaired two-tailed Student's t-test, where $p<0.0001$ is indicated by ****.

The online version of this article includes the following source data and figure supplement(s) for figure 3:

**Source data 1.** Raw FACS files for *Figure 3H–I* (OPP uptake [LPS +IL-4]).

**Figure supplement 1.** Protein synthesis is increased in B cells regardless of stimuli.

**Figure supplement 1—source data 1.** Raw FACS files for *Figure 3—figure supplement 1A–B* (OPP uptake [all agonists]).

Increased synthesis alone cannot satisfy the increased need for amino acids in proliferating cells, as the cell is unable to make essential amino acids. Several plasma membrane amino acid transporters were detected in the proteomic analysis, and these were all found to be upregulated in response to stimulation with LPS + IL-4 (*Figure 4A*). To look at the requirement for increased amino acid uptake during B cell proliferation more closely, we focused on Lat1, a member of the System L transporters. Lat1 is a heterodimer composed of an amino acid transporting light chain subunit, SLC7A5, and a chaperone heavy chain subunit, SLC3A2 (CD98) (*Napolitano et al., 2015*). Both of these subunits were upregulated in the proteomic data following LPS + IL-4 stimulation (*Figure 4B and C*). Additionally, the upregulation of SLC3A2 was confirmed by flow cytometry (*Figure 4D and E*). To determine whether this increased protein expression resulted from increased transcription of the relevant genes, we analysed RNA-Seq data generated from short-term LPS stimulation of murine B cells (*Tesi et al., 2019*). Both SLC7A5 and SLC3A2 mRNA expression was significantly upregulated following LPS stimulation (*Figure 4—figure supplement 2A–B*). A similar study using anti-IgM and anti-CD40 to activate murine B cells has found an upregulation of amino acid transporters, including SLC7A5, in their proteomic data, suggesting that this is not a stimulus-specific effect (*James et al., 2026*).

*Slc7a5*<sup>fl/fl</sup>*/Tg(Vav1-iCre)*<sup>+/-</sup> mice, which have a deletion of *Slc7a5* in all hematopoietic cells, have a comparable number of naive B cells to wild-type (WT) controls, indicating that SLC7A5 is not critical for initial B cell development (*Sinclair et al., 2013*). To examine how the loss of SLC7A5 affects B cell proliferation in response to LPS + IL-4 stimulation, B cells were purified from WT and *Slc7a5*<sup>fl/fl</sup>*/Tg(Vav1-iCre)*<sup>+/-</sup> mice, labelled with CTV, and stimulated for 72 hr with LPS + IL-4 ex vivo. The percentage of viable B cells after 72 hr was much lower in *Slc7a5*<sup>fl/fl</sup>*/Tg(Vav1-iCre)*<sup>+/-</sup> mice compared to WT B cells (*Figure 4F*). Similarly, while WT B cells proliferated, the majority of the *Slc7a5*<sup>fl/fl</sup>*/Tg(Vav1-iCre)*<sup>+/-</sup> B cells failed to divide, and the number of live B cells after 72 hr was much lower compared to WT B cells (*Figure 4G and H*). In line with the failure to proliferate, there was no evidence of class switching in the remaining *Slc7a5*<sup>fl/fl</sup>*/Tg(Vav1-iCre)*<sup>+/-</sup> B cells (*Figure 4I*). SLC7A5 transports large neutral amino acids, including phenylalanine, histidine, methionine, tryptophan, and isoleucine, as well as the tryptophan metabolite kynurenine (*Fotiadis et al., 2013*). As kynurenine is fluorescent, and its uptake can be measured by flow cytometry, the effect of LPS + IL-4 stimulation on kynurenine uptake in B cells was determined. In line with the increased expression of SLC7A5 in LPS + IL-4 treated B cells, kynurenine uptake was also higher in B cells stimulated with LPS + IL-4 for 24 hr compared to naive B cells (*Figure 4J*). Kynurenine uptake in the LPS + IL-4 stimulated B cells was reduced by the competitive substrate 2-aminobicyclo-(2,2,1)-heptane-2-carboxylic acid (BCH), which inhibits uptake through System L transporters. Likewise, kynurenine uptake in *Slc7a5*<sup>fl/fl</sup>*/Tg(Vav1-iCre)*<sup>+/-</sup> B cells was similar to the uptake in BCH-treated WT B cells, suggesting that SLC7A5 was the major system L transporter in LPS + IL-4 stimulated B cells (*Figure 4J*). BCH or SLC7A5 knockout did not reduce kynurenine uptake to baseline levels, indicating that other transport mechanisms for kynurenine uptake exist in B cells. Similar results were obtained for B cells stimulated with LPS alone, suggesting that LPS rather than IL-4 is the main driver of kynurenine uptake (*Figure 4K*).

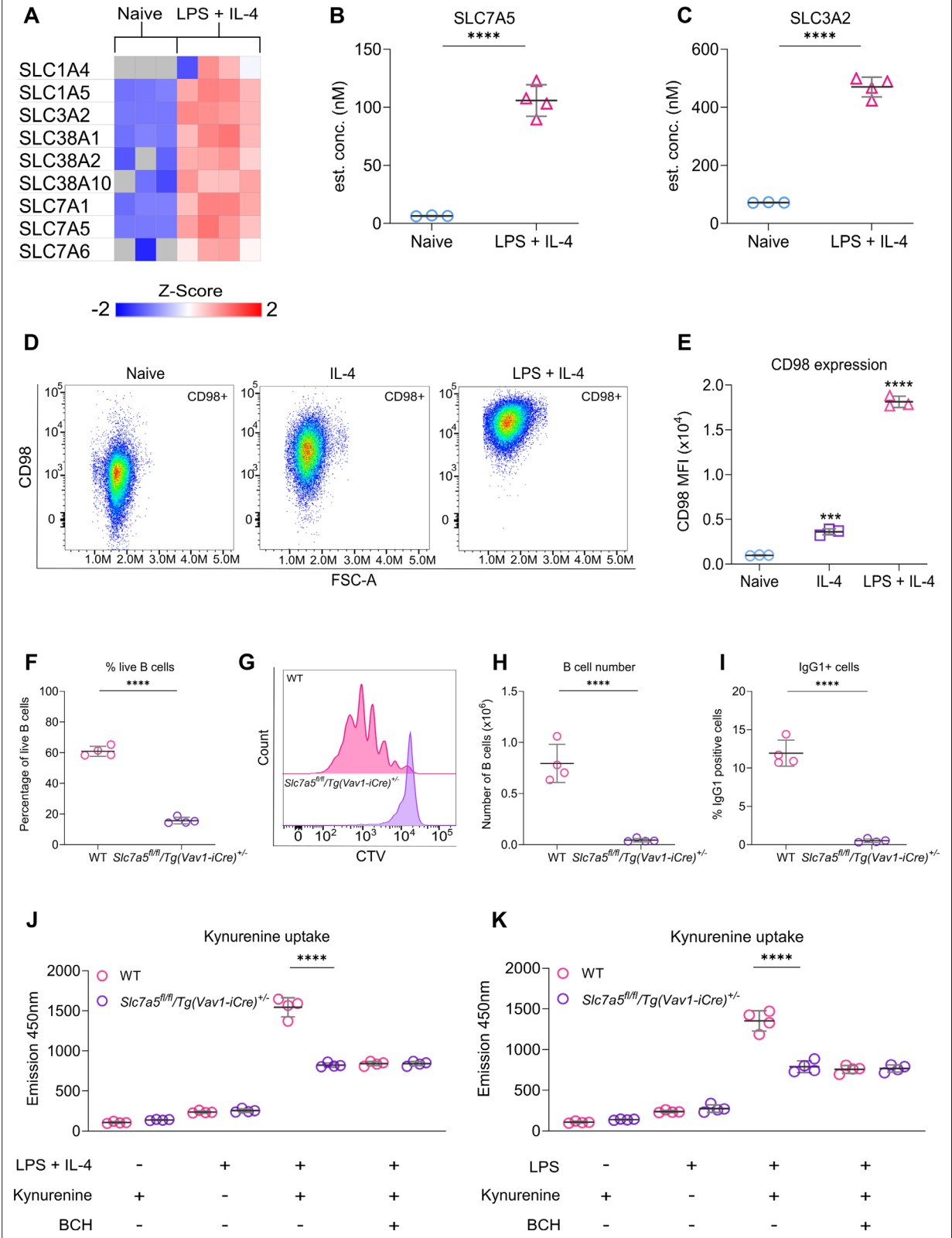

**Figure 4.** Amino acid transporter SLC7A5 is required for key B cell functions. (**A**) Heat map showing the expression of genes encoding for proteins involved in plasma membrane amino acid transport determined from the proteomic dataset described in *Figure 1*. (**B, C**) Graphs depicting changes in cellular concentration (nM) of (**B**) SLC7A5 and (**C**) SLC3A2 derived from the proteomic data. Adjusted p values from the FDR calculations applied to the proteomic dataset, where *p(adj)*<0.0001 is indicated by ****. (**D–E**) Lymph node cells from C57BL6/J mice were stimulated with LPS (20 μg/

*Figure 4 continued*

ml) and IL-4 (10 ng/ml) for 24 hr before staining for CD98. (**D**) Representative FACS plot comparing CD98 expression. (**E**) Quantification shows three technical replicates from cells isolated from one mouse and is representative of two independent experiments. Statistical power was determined using one-way ANOVA followed by multiple comparison testing via Dunnett's analysis. For comparison to naive B cells, where $p<0.001$ is indicated by *** and $p<0.0001$ by ****. (**F–I**) B cells were purified from the spleens of wild type (WT) and *Slc7a5$^{fl/fl}$/Tg(Vav1-iCre)$^{+/-}$* mice and stained with CTV before stimulation with LPS (20 µg/ml) and IL-4 (10 ng/ml) for 72 hr. (**F**) Percentage of live B cells (7AAD-ve) in WT and *Slc7a5$^{fl/fl}$/Tg(Vav1-iCre)$^{+/-}$* mice. (**G**) Histogram representing CTV staining of WT and *Slc7a5$^{fl/fl}$/Tg(Vav1-iCre)$^{+/-}$* B cells. (**H**) Live B cell number of WT and *Slc7a5$^{fl/fl}$/Tg(Vav1-iCre)$^{+/-}$* mice. (**I**) Percentage of B cells that are IgG1+ve in WT and *Slc7a5$^{fl/fl}$/Tg(Vav1-iCre)$^{+/-}$* mice. Data shows the results of four biological replicates per genotype. Statistical power was determined using an unpaired two-tailed Student's t-test, where $p<0.0001$ is indicated by ****. (**J–K**) B cells were purified from the spleens of WT and *Slc7a5$^{fl/fl}$/Tg(Vav1-iCre)$^{+/-}$* mice and stimulated with LPS (20 µg/ml) +/- IL-4 (10 ng/ml) for 24 hr before fixing and staining for the uptake of kynurenine to measure amino acid uptake. Gating strategy for kynurenine uptake in ***Figure 4—figure supplement 3A***. (**J**) Quantification of kynurenine MFI between B cells from WT and *Slc7a5$^{fl/fl}$/Tg(Vav1-iCre)$^{+/-}$* mice with (+) or without (-) LPS +IL-4, kynurenine or aminobicyclo-(2,2,1)-heptane-2-carboxylic acid (BCH). (**K**) Quantification of kynurenine MFI between B cells from WT and *Slc7a5$^{fl/fl}$/Tg(Vav1-iCre)$^{+/-}$* mice with (+) or without (-) LPS, kynurenine or BCH. Data shows the results of four biological replicates per genotype. Statistical power was determined for using two-way ANOVA followed by multiple comparison testing via Sidak's analysis, where $p<0.0001$ is indicated by **** for comparisons between genotypes.

The online version of this article includes the following source data and figure supplement(s) for figure 4:

**Source data 1.** Raw FACS files for ***Figure 4D–E*** (CD98 staining), 4F-I (WT vs SLC7A5 KO proliferation).

**Source data 2.** Raw FACS files for ***Figure 4J*** (Kynurenine uptake – LPS +IL-4 [WT vs SLC7A5 KO]), 4 K (Kynurenine uptake – LPS [WT vs SLC7A5 KO]).

**Figure supplement 1.** LPS + IL-4 stimulation upregulates serine biosynthesis.

**Figure supplement 2.** Short-term LPS stimulation increases the transcription of genes involved in amino acid uptake and cholesterol metabolism in B cells.

**Figure supplement 3.** Representative gating strategies.

**Figure supplement 3—source data 1.** Raw FACS files for ***Figure 4—figure supplement 3A*** (Kynurenine uptake gating strategy), 3B (filipin staining gating strategy), 3 C (purified B cell gating strategy).

## Cholesterol metabolism is upregulated in activated B cells

The enrichment analysis (***Figure 1H***) suggested that sterol biosynthesis may be upregulated by LPS + IL-4 stimulation. Sterol biosynthesis allows the generation of cholesterol from acetyl-CoA within the cell via a complex multistep process (***Figure 5—figure supplement 1***). The 1st half of this pathway, often referred to as the mevalonate pathway, results in the production of farnesyl pyrophosphate (***Buhaescu and Izzedine, 2007***). This can be used for several purposes in the cell, including the synthesis of cholesterol via the Bloch or Kandutsch-Russell pathways (***Mitsche et al., 2015***), prompting us to investigate this pathway further. Analysis of the proteomic data showed that the majority of enzymes involved in cholesterol biosynthesis were increased in LPS + IL-4 stimulated B cells relative to naive B cells (***Figure 5A***). Cholesterol biosynthesis is controlled by two rate-limiting steps: the conversion of HMG-CoA to mevalonic acid, which is catalysed by HMG-CoA-reductase (HMGCR), and the conversion of squalene to 2,3(*S*)-oxidosqualene, catalysed by squalene monooxygenase (SQLE; ***Buhaescu and Izzedine, 2007***). While neither of these enzymes were detected in the proteomic dataset for naive B cells, they were both consistently expressed after LPS + IL-4 stimulation (***Figure 5B and C***). Cells can also obtain cholesterol via the uptake of low-density lipoprotein (LDL) through the low-density lipoprotein receptor (LDLR) (***Luo et al., 2020***). Similar to HMGCR and SQLE, the LDLR was not found in naive B cells but was detected in B cells following LPS + IL-4 stimulation (***Figure 5D***). Analysis of published transcriptomic data (***Tesi et al., 2019***) on LPS stimulated B cells showed that *Hmgcr*, *Sqle*, and *Ldlr* mRNA expression was significantly increased by LPS stimulation and peaked after 4 hr of stimulation (***Figure 4—figure supplement 2C–E***).

Cholesterol homeostasis is regulated by the transcription factor SREBP2, which is found as an immature precursor protein embedded in the endoplasmic reticulum (ER) with chaperones SCAP and INSIG (***Luo et al., 2020***; ***Brown et al., 2018***). Low sterol levels are detected by the sterol-sensing domain of SCAP, leading to the dissociation and degradation of INSIG. Subsequently, the SCAP-SREBP2 complex is transported from the ER to the Golgi apparatus, where it interacts with site 1 and site 2 proteases, allowing cleavage of SREBP2. The mature form of SREBP2 then translocates to the nucleus, resulting in the upregulation of genes involved in cholesterol metabolism, including *hmgcr*, *sqle*, and *ldlr* (***Luo et al., 2020***; ***Brown et al., 2018***). Low levels of SREBP2 expression were detectable in naive B cells and became upregulated following LPS + IL-4 stimulation (***Figure 5E***). Conversely,

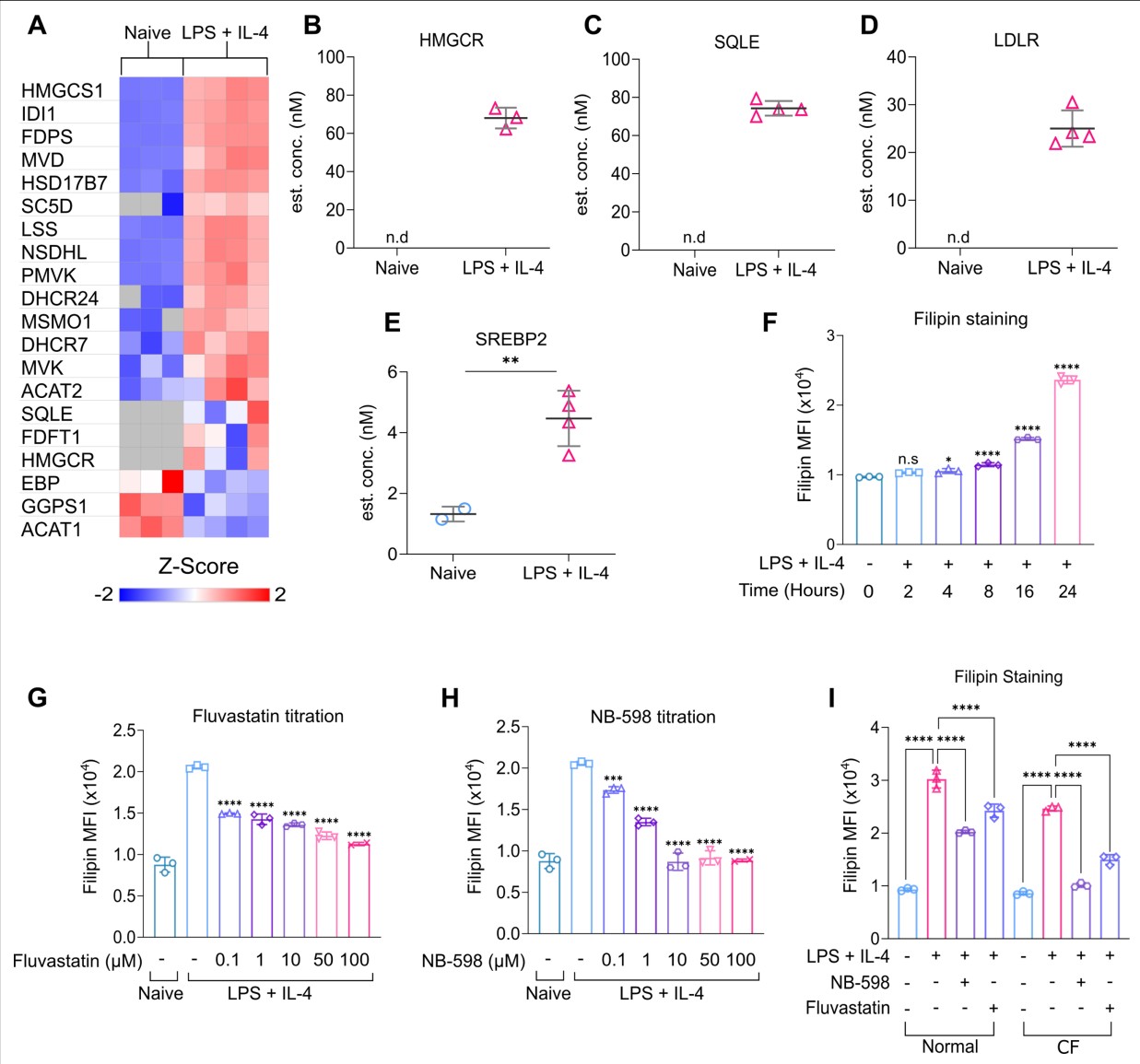

**Figure 5.** LPS + IL-4 stimulation upregulates cholesterol metabolism in B cells. (**A–E**) Analysis of the genes involved in cholesterol metabolism in the proteomic dataset. (**A**) Heat map of all the enzymes involved in cholesterol biosynthesis. Cellular concentration (nM) of (**B**) HMGCR, (**C**) SQLE, (**D**) LDLR, and (**E**) SREBP2. Adjusted p values from the FDR calculations applied to the proteomic dataset, where *p(adj)*<0.01 is indicated by **. (**F**) Splenocytes from C57BL6/J mice were plated in cholesterol-free (CF) media and stimulated with LPS (20 µg/ml) and IL-4 (10 ng/ml). The cells were fixed at the stated time points and stained with filipin. The gating strategy for filipin staining in *Figure 4—figure supplement 3B*. (**F**) Filipin time course. Data shows three technical replicates from cells isolated from one mouse. Statistical power was determined by one-way ANOVA followed by multiple comparison testing via Dunnett's analysis. For comparisons to naive B cells, p<0.05 is indicated by * and p<0.0001 by ****. ns indicated by p>0.05. (**G–H**) Splenocytes from C57BL6/J mice were plated in cholesterol-free media and pre-treated with DMSO as a vehicle control or varying concentrations of Fluvastatin or NB-598 for 45 min before stimulation with LPS (20 µg/ml) and IL-4 (10 ng/ml). The cells were fixed after 24 hr and stained with filipin before acquisition. (**G**) Filipin staining of Fluvastatin titration. (**H**) Filipin staining of NB-598 titration. Data shows three technical replicates from cells isolated from one mouse and is representative of two independent experiments, each with one biological replicate. Statistical power was determined using a one-way ANOVA followed by multiple comparison testing via Dunnett's analysis. For comparison to LPS + IL-4 stimulated B cells, p<0.001 is indicated by *** and p<0.0001 by ****. (**I**) Splenocytes from C57BL6/J mice were plated in normal or cholesterol-free (CF) media and pre-treated with DMSO as a vehicle control, Fluvastatin (10 µM) or NB-598 (10 µM) before stimulation with LPS +IL-4. The cells were fixed after 24 hr and stained with filipin before acquisition. Data shows the results of three biological replicates. Statistical power was determined using two-way ANOVA followed by multiple comparison testing via Tukey's analysis. For comparisons to the LPS + IL-4 condition, p<0.0001 is indicated by ****.

The online version of this article includes the following source data and figure supplement(s) for figure 5:

**Source data 1.** Raw FACS files for *Figure 5F* (filipin staining timecourse), 5 G (filipin staining – Fluvastatin titration).

*Figure 5 continued on next page*

*Figure 5 continued*

**Source data 2.** Raw FACS files for *Figure 5H* (filipin staining – NB-598 titration), 5I (filipin staining – Fluvastatin/NB-598 treatment (normal/CF)).

**Figure supplement 1.** Terpenoid and cholesterol biosynthetic pathway.

**Figure supplement 2.** Blocking rate-limiting enzymes in the cholesterol biosynthesis pathway reduces B cell growth, survival, and proliferation.

**Figure supplement 2—source data 1.** Raw FACS files for *Figure 5—figure supplement 2A–C* (Fluvastatin titration – 24 hr LPS + IL-4 stimulation), 2D-F (NB-598 titration – 24 hr LPS + IL-4 stimulation).

**Figure supplement 2—source data 2.** Raw FACS files for *Figure 5—figure supplement 2G–I* (Rosuvastatin titration – 24 hr LPS + IL-4 stimulation), 2 J (filipin staining – Rosuvastatin titration).

there was no significant difference in the mRNA expression of *Srebp2* (*Figure 4—figure supplement 2F*), suggesting that the regulation of SREBP2 levels may occur post-transcriptionally.

Together, the above results suggest that LPS + IL-4 stimulation would drive an increase in cellular cholesterol. To test this, B cells were stained with the fluorescent dye filipin, which binds to cellular cholesterol. LPS + IL-4 stimulation resulted in a gradual increase in cholesterol levels from 4 hr to 24 hr of stimulation (*Figure 5F*). As cells can obtain cholesterol via de novo synthesis or uptake of LDL from serum, we compared the effects of culturing B cells in normal serum or serum treated with fumed silica to remove LDL (*Brovkovych et al., 2019*). Analysis of the serum indicated that this treatment successfully removed LDL and HDL but did not affect the levels of triglycerides (*Supplementary file 2*). Culture in media free of LDL (cholesterol-free (CF) media) did not prevent the increase in filipin staining stimulated by LPS + IL-4 (*Figure 5G–I*). Inhibitors of cholesterol biosynthesis were titrated to determine what concentration was required to reduce filipin staining in B cells stimulated in CF media. The increase in cholesterol following LPS + IL-4 stimulation could be reduced by the HMGCR inhibitor Fluvastatin or by NB-598, an inhibitor of SQLE (*Figure 5G and H*). Cells remained viable after 24 hr of inhibitor treatment with up to 100 µM of either inhibitor (*Figure 5—figure supplement 2A–F*). Furthermore, the ability of HMGCR/SQLE inhibition to reduce cholesterol levels did not correlate with effects on cell size, as judged by FSC measurements by flow cytometry after 24 hr (*Figure 5—figure supplement 2B,E*). From this, a concentration of 10µM was chosen for both NB-598 and Fluvastatin, as this was sufficient to reduce cholesterol as close to naive B cell levels as possible, but did not cause significant cell death after 24 hr. Different classes of statins have been described to exert different pleiotropic effects in vivo (*Greenwood et al., 2006*; *Oesterle et al., 2017*). Depending on their solubility, statins can be categorised as hydrophilic or lipophilic. Lipophilic statins, such as Fluvastatin, passively diffuse into cells, whereas hydrophilic statins may enter the cell through active transport. In the case of LPS + IL-4 stimulated B cells, the use of a more hydrophilic statin, Rosuvastatin, gave similar results to Fluvastatin (*Figure 5—figure supplement 2G–J*).

The increase in filipin staining following LPS + IL-4 stimulation for 24 hr in normal media was reduced, but not abolished, by either NB-598 or Fluvastatin treatment (*Figure 5I*). The ability of NB-598 and Fluvastatin treatment to reduce LPS + IL-4 induced filipin staining was greater in CF media relative to normal media that contains cholesterol (*Figure 5I*). Together, these results suggest that B cells may obtain cholesterol via a combination of biosynthesis and uptake.

## The mevalonate pathway is critical for the survival and proliferation of LPS + IL-4 stimulated B cells

As the above data indicated that B cells increased their cholesterol levels following LPS + IL-4 stimulation, we assessed the importance of this process for B cell proliferation ex vivo.

Stimulation with LPS + IL-4 for 48 hr in normal media promoted proliferation, and this was reduced by treatment with either Fluvastatin or NB-598 (as shown by CTV staining, *Figure 6A*). In line with this decreased proliferation, both Fluvastatin and NB-598 greatly reduced the number of B cells present at 48 hr compared to cells treated with LPS + IL-4 alone (*Figure 6B*). The remaining cells in the Fluvastatin or NB-598 treated conditions also showed significantly lower levels of survival (based on 7-AAD staining, *Figure 6C*) and failed to increase in size (as judged by forward scatter, *Figure 6D*). Similarly, B cells grown in CF media were still able to proliferate at a normal rate and experienced similar inhibitory effects after treatment with Fluvastatin or NB-598 (*Figure 6A–D*).

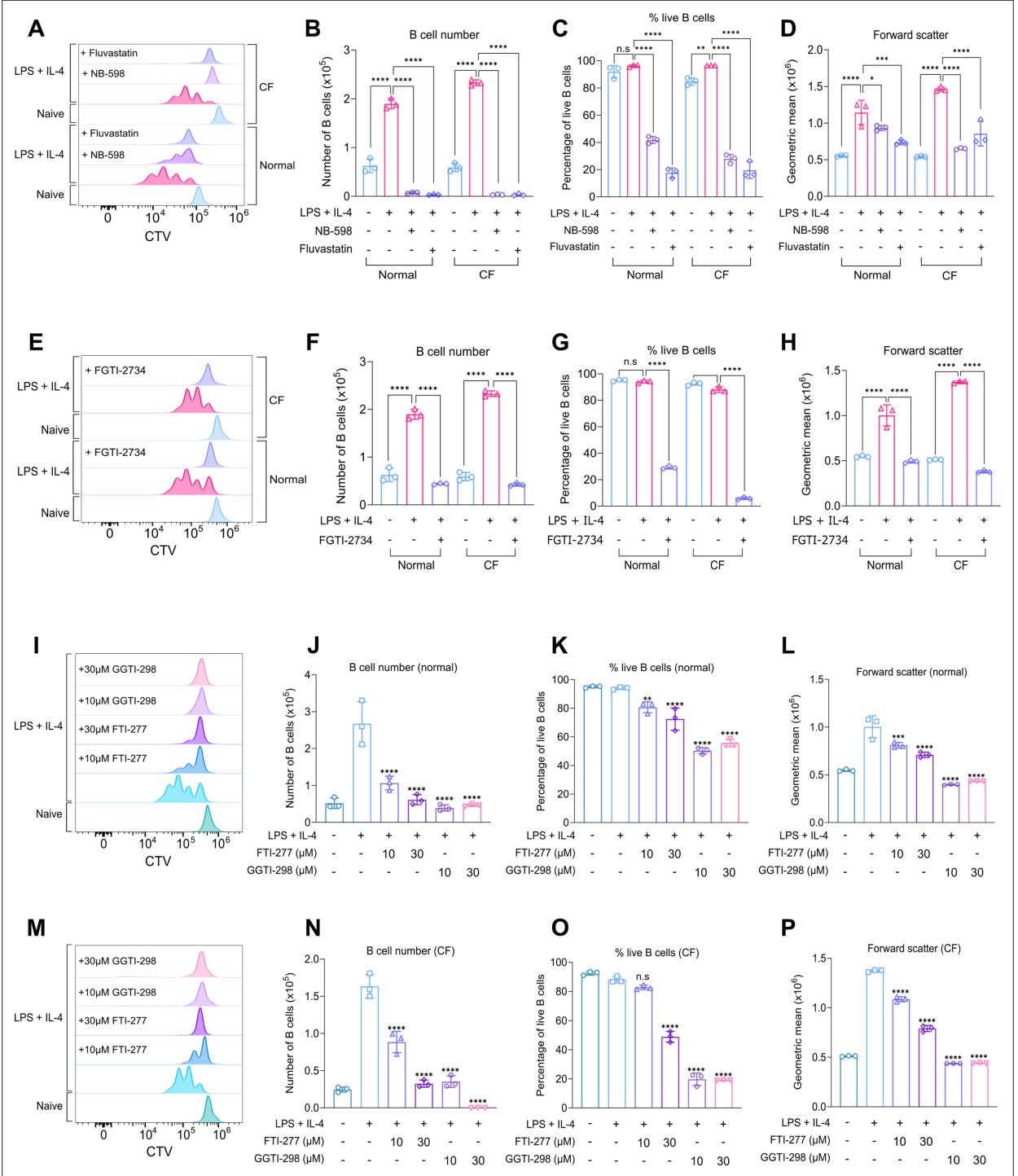

**Figure 6.** Blocking rate-limiting enzymes in the cholesterol biosynthesis pathway reduces B cell growth, survival, and proliferation. (**A–D**) B cells were purified from the spleens of C57BL6/J mice and cultured in normal or cholesterol-free (CF) media. The cells were stained with CTV, then pre-treated with DMSO as a vehicle control, Fluvastatin (10 μM), or NB-598 (10 μM), where indicated, for 45 min prior to stimulation with LPS (20 μg/ml) and IL-4 (10 μg/ml) for 48 hr. Gating strategy for CTV staining is shown in *Figure 4—figure supplement 3C*. (**A**) shows representative CTV staining, (**B**) live B cell number, (**C**) percentage of live B cells (7AAD-ve), and (**D**) forward scatter. Data shows the results of three biological replicates. (**E–H**) Same as (**A–D**) but with pretreatment using FGTI-2734 (10 μM). (**E**) shows representative CTV staining, (**F**) live B cell number, (**G**) percentage of live B cells (7AAD-ve), and (**H**) forward scatter of B cells. Data shows the results of three biological replicates. (**I–P**) Same as (**A–D**) but with pretreatment using FTI-277 or GGTI-298 (10 μM-30 μM). (**I**) shows representative CTV staining in normal media, (**J**) live B cell number, (**K**) percentage of live B cells (7AAD-ve), and (**L**) forward scatter of B cells. (**M**) shows representative CTV staining in CF media, (**N**) live B cell number, (**O**) percentage of live B cells (7AAD-ve), and (**P**) forward

*Figure 6 continued on next page*

*Figure 6 continued*

scatter of B cells. Data shows the results of three biological replicates. (**B–D**) (**F–H**) Statistical power was determined using two-way ANOVA followed by multiple comparison testing via Tukey's analysis. For comparisons to the LPS + IL-4 condition, $p < 0.05$ is indicated by *, $p < 0.01$ by **, $p < 0.001$ by *** and $p < 0.0001$ by ****. ns indicated by $p > 0.05$. (**J–L**) (**N–P**) Statistical power was determined using one-way ANOVA followed by multiple comparison testing via Dunnett's analysis. For comparisons to the LPS + IL-4 condition, $p < 0.01$ is indicated by **, $p < 0.001$ by *** and $p < 0.0001$ by ****. ns indicated by $p > 0.05$. For all panels, cells in the absence of LPS + IL-4 were naive B cells analysed on the day of isolation.

The online version of this article includes the following source data and figure supplement(s) for figure 6:

**Source data 1.** Raw FACS files for *Figure 6A–D* (proliferation – Fluvastatin/NB-598 treatment [normal/CF]), 6E-H (proliferation – FGTI-2734 treatment [normal/CF]).

**Source data 2.** Raw FACS files for *Figure 6I–L* (GGTI-298 treatment [normal/CF]), 6 M-P (proliferation – FTI-277 [normal/CF]).

**Figure supplement 1.** Blocking cholesterol biosynthesis or uptake upregulates proteins involved in cholesterol metabolism in B cells.

To determine if blocking cholesterol biosynthesis or cholesterol uptake induced any major changes in the proteome, we analysed splenic B cells after 24 hr of LPS + IL-4 stimulation in normal and CF media in the presence or absence of Fluvastatin, using DIA-based mass spectrometry.

Using cutoffs of a log2 fold change more than one standard deviation away from the median and $q < 0.05$, no significant changes in protein expression were found when comparing LPS + IL-4 stimulated B cells cultured in normal media to those in CF media (*Figure 6—figure supplement 1A*). There was an upregulation of 31 proteins and a downregulation of 12 proteins when comparing LPS + IL-4 stimulated B cells grown in normal media with or without Fluvastatin (*Figure 6—figure supplement 1B*). The enrichment analysis (*Supplementary file 1*) did not highlight any specific processes in the downregulated proteins, but the upregulated proteins were enriched in terms associated with steroid biosynthesis. This suggests Fluvastatin treatment does not cause any major off-target effects in these B cells. Comparing LPS + IL-4 stimulated B cells grown in normal media to those in CF media + Fluvastatin, there was an upregulation of 32 proteins and downregulation of 35 proteins (*Figure 6—figure supplement 1C*). Again, the enrichment analysis (*Supplementary file 1*) did not highlight any specific processes in the downregulated proteins, but there was an enrichment for proteins involved in steroid biosynthesis in the upregulated proteins. Comparing upregulated proteins from *Figure 6—figure supplement 1B and C* across all conditions indicated that, for most changes, the combination of CF media and Fluvastatin had the greatest effect (*Figure 6—figure supplement 1D*). A similar pattern was observed for the downregulated proteins (*Figure 6—figure supplement 1E*). Looking specifically at proteins involved in cholesterol metabolism, the majority of these were upregulated by treatment with Fluvastatin or growth in CF media (*Figure 6—figure supplement 1F*), including the rate-limiting enzymes HMGCR and SQLE (*Figure 6—figure supplement 1G, H*). In addition, there was upregulation of the LDLR (*Figure 6—figure supplement 1I*). Again, the biggest effect on these proteins was seen with the combination of CF media and Fluvastatin.

In addition to cholesterol biosynthesis, the mevalonate pathway is also required for protein prenylation, a process that involves the addition of a farnesyl or geranylgeranyl motif to cysteine residues near the C-terminus of target proteins (*Figure 5—figure supplement 1*; *Wang and Casey, 2016*). As SQLE acts downstream of this branch point, the effects of NB-598 would argue for an important role of cholesterol biosynthesis in LPS + IL-4 induced proliferation; however, they would not exclude a parallel role for protein prenylation. To determine if the inhibitory effects of Fluvastatin in stimulated B cells may be in part due to changes in prenylation, B cells were treated with a dual prenylation inhibitor, FGTI-2734 (*Kazi et al., 2019*), which blocks the enzymes responsible for catalysing prenylation: farnesyl transferase (FTase) and geranylgeranyl transferase (GGTase). Treatment of B cells with FGTI-2734 for 48 hr decreased proliferation in response to LPS + IL-4 stimulation in normal media (*Figure 6E*), and in line with this, the number of live B cells also decreased (*Figure 6F*). Cell survival was significantly reduced to the same level as seen with Fluvastatin treatment (compare *Figure 6G–C*). FGTI-2734 also had a major impact on cell size in normal media (*Figure 6H*). Similar effects were observed in CF media (*Figure 6E–H*). To determine whether inhibiting farnesylation or geranylgeranylation was responsible for the inhibitory effect of FGTI-2734, B cells were treated with the FTase inhibitor FTI-277 (*Lerner et al., 1995*) or the GGTase inhibitor GGTI-298 (*McGuire et al., 1996*). In normal media, treatment with 10 µM of FTI-277 reduced proliferation to around two generations of B cells (*Figure 6I*), whereas 30 µM of FTI-277 had a more significant effect on proliferation (*Figure 6I*). Inhibition of GGTase with either 10 µM or 30 µM of GGTI-298 blocked

proliferation entirely (*Figure 6I*). In line with this, the number of live B cells was significantly reduced by treatment with either inhibitor (*Figure 6J*). In terms of survival, FTI-277 minimally reduced the percentage of live B cells, whereas GGTI-298 had a more profound impact on survival (*Figure 6K*). Similarly, treatment with FTI-277 partially reduced cell size, but GGTI-298 reduced cell size to a level comparable to naïve B cells (*Figure 6L*). Similar effects were observed in CF media (*Figure 6M–P*); however, treatment with GGTI-298 reduced survival to an even greater extent compared to normal media (*Figure 6O*).

In theory, the effects of HMGCR inhibition should be reduced if cells were supplemented with a metabolite downstream of HMGCR in the mevalonate pathway. To test this, the effects of supplementing growth medium with mevalonate, the product generated in the reaction catalysed by HMGCR, were tested. For this, a concentration of 2 mM mevalonate was used, as initial experiments showed that higher concentrations resulted in cell death and reduced filipin staining (*Figure 7—figure supplement 1*). Initially, the ability of mevalonate to rescue the effects of Fluvastatin on filipin staining was examined. In cells treated for 48 hr with LPS + IL-4, the reduction in filipin caused by Fluvastatin treatment in normal media was reversible by the addition of mevalonate (*Figure 7A*). Mevalonate had less effect on cells stimulated for 24 hr with LPS + IL-4, although at this time point, the ability of Fluvastatin to reduce filipin staining was also less pronounced than at 48 hr (*Figure 7A*). Surprisingly, in CF media, mevalonate was unable to rescue the effects of Fluvastatin at either time point (*Figure 7B*).

The effect of Fluvastatin on cell survival was partially reversible by the addition of mevalonate in normal media, which restored cell survival to 80% (*Figure 7C*) and fully restored cell size relative to cells stimulated with LPS and IL-4 in the absence of Fluvastatin (*Figure 7D*). However, mevalonate supplementation was not sufficient to completely rescue proliferation, as CTV staining demonstrated that 2 generations of B cells were present rather than 4 generations after LPS + IL-4 stimulation alone (*Figure 7E*). The number of live B cells following mevalonate addition was significantly higher compared to the Fluvastatin only condition, but did not reach the same level as LPS + IL-4 stimulation alone (*Figure 7F*). Similar to what was seen for filipin staining, mevalonate was unable to rescue any of the inhibitory effects of Fluvastatin treatment in CF media (*Figure 7C–F*).

Since inhibiting geranylgeranyl transferase had such a profound effect on B cell function, the role of prenylation was also tested by supplementing cells with geranylgeranyl pyrophosphate (GGPP), which can be used for prenylation but not cholesterol biosynthesis. Surprisingly, GGPP was more effective than mevalonate in rescuing the effect of Fluvastatin on LPS + IL-4 induced proliferation (*Figure 8A*), and in line with this, there was a greater number of live B cells in the GGPP-treated condition relative to the mevalonate-treated condition (*Figure 8B*). A combination of GGPP and mevalonate did not have a synergistic effect compared to GGPP alone (*Figure 8A and B*). GGPP was also able to rescue the effects of Fluvastatin on both cell survival and cell size (*Figure 8C and D*). Conversely, in CF media, the addition of GGPP was unable to rescue any of the inhibitory effects of Fluvastatin on cell proliferation or survival (*Figure 8E–H*).

Given that GGPP should rescue prenylation but not cholesterol metabolism, its effects on proliferation were unexpected if B cells need cholesterol to proliferate. We therefore tested if GGPP affected cholesterol levels in Fluvastatin-treated B cells. As seen in previous experiments, Fluvastatin reduced cholesterol levels at both 24 hr and 48 hr of treatment. GGPP was able to partially rescue this at both time points (*Figure 8I*). This was in contrast to mevalonate, which could only rescue at 48 hr (*Figure 7A*), an observation that may, in part, explain why GGPP was more effective at restoring proliferation in the presence of Fluvastatin than mevalonate. The effects of GGPP on cholesterol levels were, however, limited to normal media, as similar to mevalonate, GGPP did not rescue the effect of Fluvastatin on filipin staining in CF media (*Figure 8J*).

## MAPK and mTOR signalling are required for B cell proliferation and cholesterol metabolism

To investigate how LPS induces these metabolic changes, we examined signalling pathways activated downstream of TLR4. MyD88 is a key adaptor protein which links most TLRs, including TLR4, to downstream signalling proteins (*Kawai et al., 2024*). To confirm the necessity of this protein for LPS signalling, we stimulated B cells from WT and MyD88[-/-] (MyD88 KO) mice (*Adachi et al., 1998*) with LPS for 30 min before immunoblotting for the phosphorylation of downstream proteins. *Myd88* deficiency prevented the phosphorylation of p38α and ERK1/2. LPS also activates PI3K/mTOR pathways

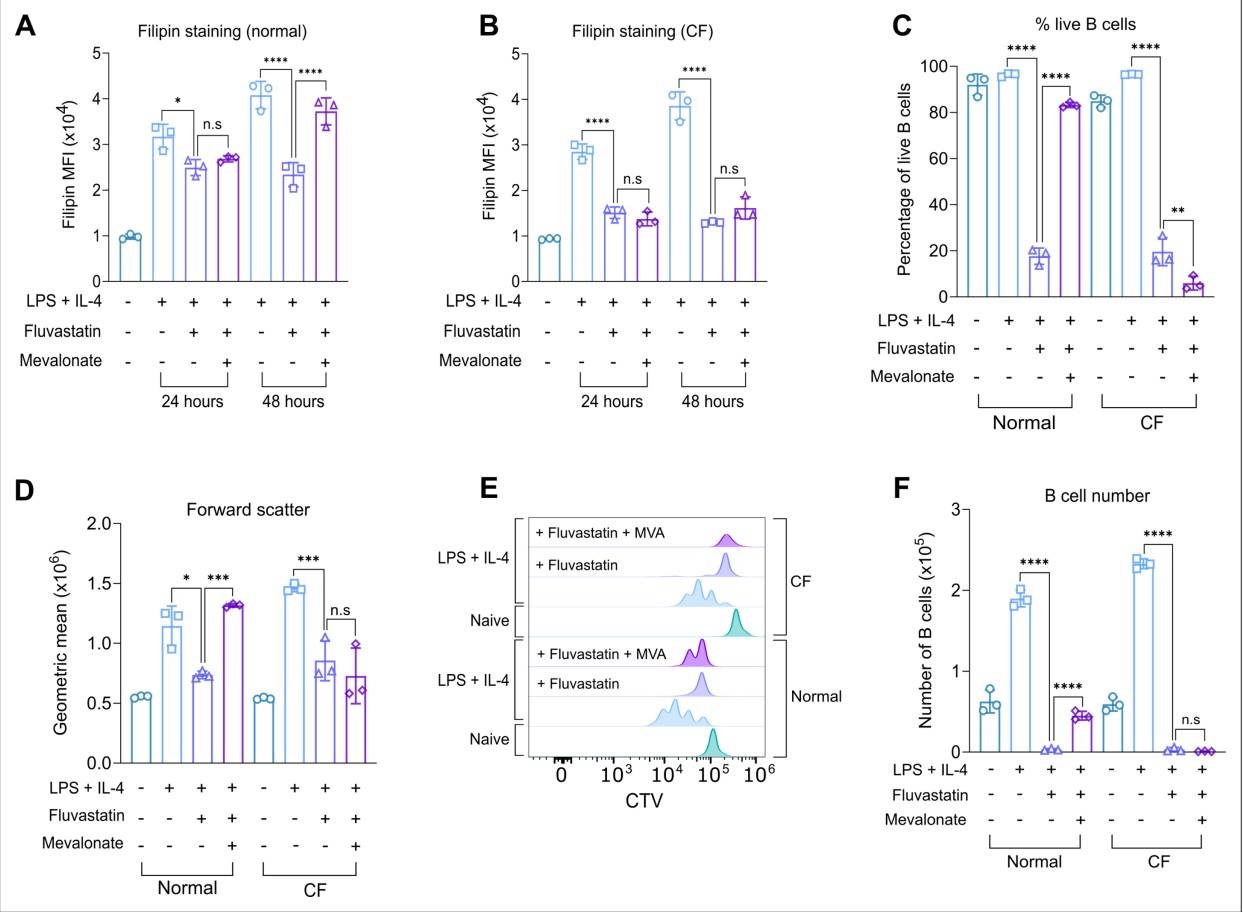

**Figure 7.** Mevalonate supplementation rescues the effect of Fluvastatin treatment in B cells. (**A–B**) Splenocytes from C57BL6/J mice were plated in normal or cholesterol-free (CF) media and pre-treated with HEPES as a vehicle control or mevalonate (2 mM) for 1 hr prior to treatment with DMSO or Fluvastatin (10 µM), where indicated, for 45 min before stimulation with LPS (20 µg/ml) and IL-4 (10 ng/ml). The cells were fixed after 24 hr or 48 hr of LPS + IL-4 stimulation and stained with filipin. (**A**) Filipin staining comparing B cells +/- Fluvastatin or mevalonate after 24 or 48 hr of LPS + IL-4 stimulation in normal media. (**B**) Filipin staining comparing B cells +/-Fluvastatin or mevalonate after 24 or 48 hr of LPS + IL-4 stimulation in CF media. Data shows the results of three biological replicates. (**C–F**) B cells were purified from the spleens of C57BL6/J mice and cultured in normal or cholesterol-free (CF) media. The cells were stained with CTV, then pre-treated with HEPES as a vehicle control or mevalonate (2 mM) for 1 hr. The cells were treated with DMSO or Fluvastatin (10 µM), where indicated, for 45 min prior to stimulation with LPS (20 µg/ml) and IL-4 (10 µg/ml) for 48 hr. (**C**) Shows percentage of live B cells (7AAD-ve), (**D**) forward scatter of B cells, (**E**) representative CTV staining, and (**F**) live B cell number. Data shows the results of three biological replicates. Where shown, statistical power was determined using two-way ANOVA followed by multiple comparison testing via Sidak's analysis. For comparison to Fluvastatin-treated B cells, p<0.05 is indicated by *, p<0.01 by **, p<0.001 by ***, p<0.0001 by ****. ns by >0.05. For all panels, cells in the absence of LPS + IL-4 were naive B cells analysed on the day of isolation.

The online version of this article includes the following source data and figure supplement(s) for figure 7:

**Source data 1.** Raw FACS files for *Figure 7A* (filipin staining – mevalonate supplementation [normal]), 7B (filipin staining - mevalonate supplementation [CF]).

**Source data 2.** Raw FACS files for *Figure 7C–F* (proliferation – mevalonate supplementation [normal/CF]).

**Figure supplement 1.** High levels of exogenous mevalonate are toxic to B cells.

**Figure supplement 1—source data 1.** Raw FACS files for *Figure 7—figure supplement 1A–B* (Mevalonate toxicity).

in B cells (*Iwata et al., 2017*) , and loss of *Myd88* reduced the phosphorylation of AKT, p70S6K, and S6 in these pathways (*Figure 9—figure supplement 1*).

To determine if these signalling pathways were important for LPS-induced B cell proliferation, B cells were treated with inhibitors of MAPK and mTOR signalling before stimulation with LPS + IL-4 for 48 hr. B cells treated with PD184352, an inhibitor of MEK1/2 (*Bain et al., 2007*), the upstream activators of ERK1/2, were still able to proliferate, with four generations of B cells present (*Figure 9A*). However, looking at the proportion of B cells per generation revealed that a higher percentage of

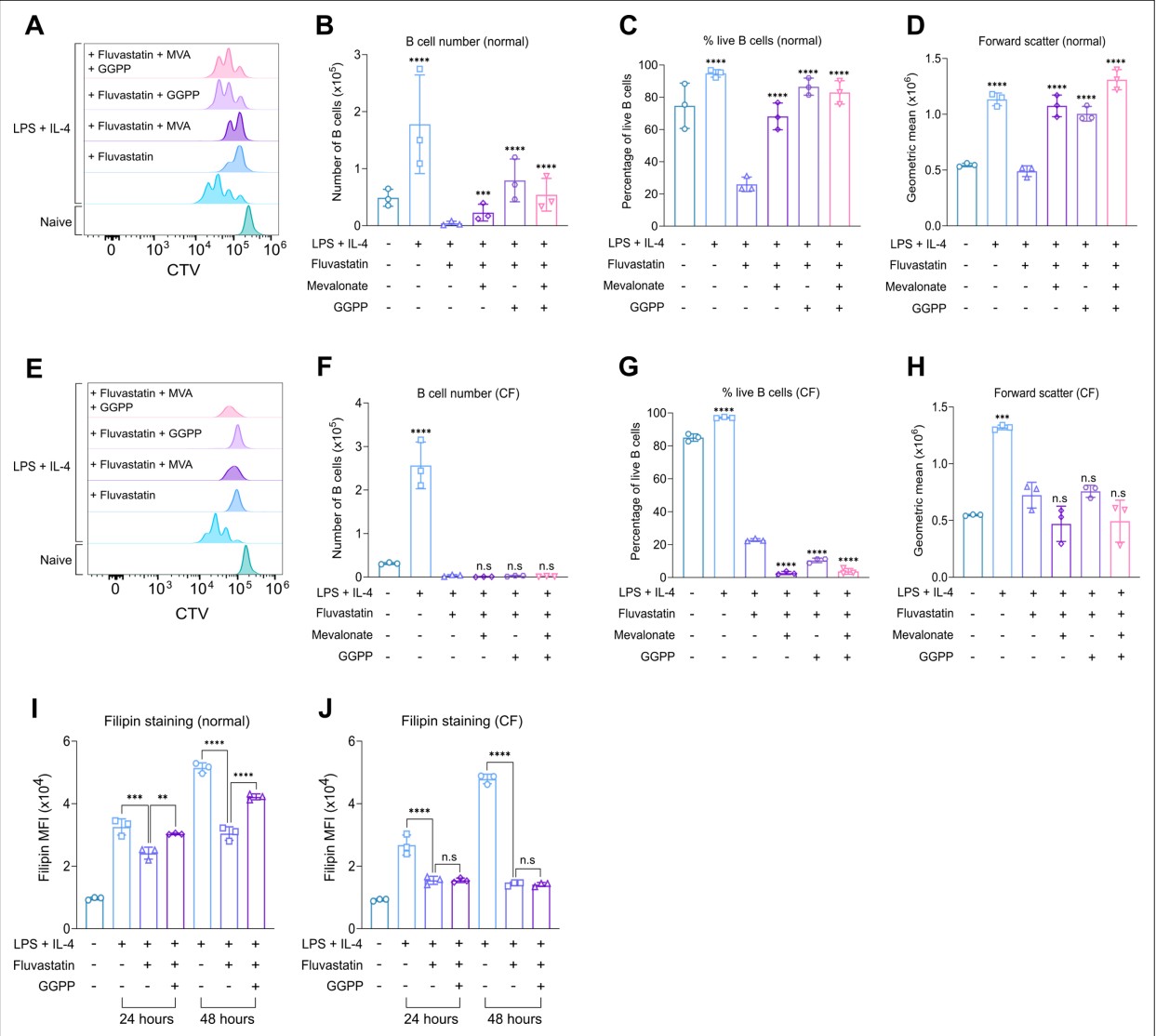

**Figure 8.** GGPP supplementation rescues the effect of Fluvastatin treatment in B cells. (**A–H**) B cells were purified from the spleens of C57BL6/J mice and cultured in normal or cholesterol-free (CF) media. The cells were stained with CTV, pre-treated with methanol:ammonium hydroxide solution (CH3OH:NH4OH) as a vehicle control, mevalonate (2 mM), or geranylgeranyl pyrophosphate (GGPP; 10 µM) for 1 hr before treatment with Fluvastatin (10 µM), where indicated. The cells were stimulated with LPS (20 µg/ml) and IL-4 (10 µg/ml) and cultured for 48 hr. (**A**) shows representative CTV staining for B cells cultured in normal media, (**B**) live B cell number, (**C**) percentage of live B cells, and (**D**) forward scatter of B cells. (**E**) shows representative CTV staining for B cells cultured in CF media, (**F**) live B cell number, (**G**) percentage of live B cells, and (**H**) forward scatter of B cells. Data shows the results of three biological replicates. Statistical power was determined using one-way ANOVA followed by multiple comparison testing via Dunnett's analysis. For comparison to Fluvastatin treated B cells, $p<0.001$ is indicated by **** and $p<0.0001$ by ****. ns indicated $p>0.05$. For (**B**) and (**F**), this data was log-transformed and then statistically analysed due to unequal variance. (**I–J**) Splenocytes were plated in normal or cholesterol-free (CF) media and pre-treated with methanol:ammonium hydroxide solution (CH3OH:NH4OH) as a vehicle control or geranylgeranyl pyrophosphate (GGPP; 10 µM), where indicated, for 1 hr before treatment with DMSO or Fluvastatin (10 µM). The cells were then stimulated with LPS (20 µg/ml) and IL-4 (10 ng/ml). The cells were fixed after 24 hr and stained with filipin before acquisition. (**I**) Filipin staining comparing B cells +/- Fluvastatin or GGPP after 24 or 48 hr of LPS + IL-4 stimulation in normal media. (**J**) Filipin staining comparing B cells +/-Fluvastatin or GGPP after 24 or 48 hr of LPS + IL-4 stimulation in CF media. Data shows the results of three biological replicates. Statistical power was determined using two-way ANOVA followed by multiple comparison testing via Sidak's analysis. For comparison to Fluvastatin-treated B cells, $p<0.01$ is indicated by **, $p<0.001$ by ***, and $p<0.0001$ by ****. ns indicated $p>0.05$. For all panels, cells in the absence of LPS + IL-4 were naive B cells analysed on the day of isolation.

The online version of this article includes the following source data for figure 8:

**Source data 1.** Raw FACS files for *Figure 8A–D* (proliferation – GGPP supplementation [normal]), 8E-H (proliferation – GGPP supplementation [CF]).

**Source data 2.** Raw FACS files for *Figure 8I* (filipin staining – GGPP supplementation [normal]), 8 J (filipin staining – GGPP supplementation [CF]).

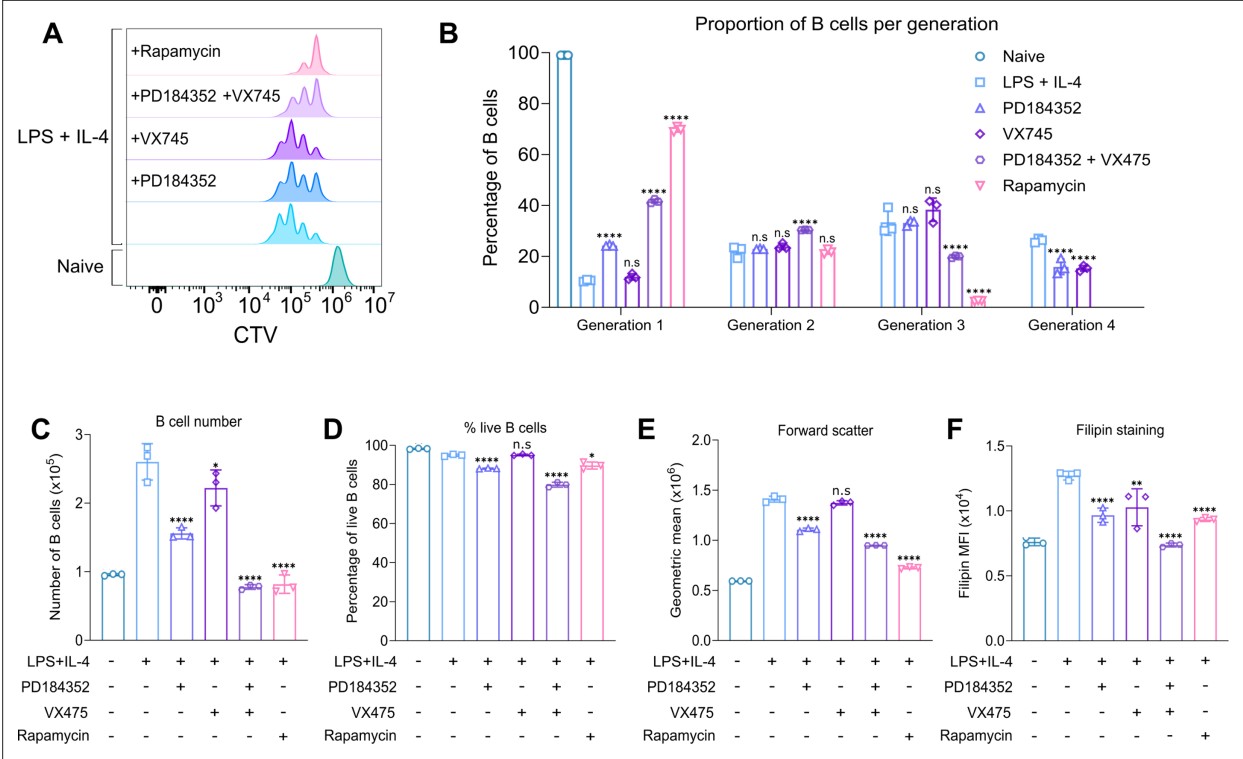

**Figure 9.** MAPK and mTOR signalling regulate B cell proliferation and cholesterol levels. (**A–E**) B cells were purified from the spleens of C57BL6/J mice and stained with CTV, then pre-treated with DMSO as a vehicle control, PD18352 (2 µM), VX745 (1 µM), or Rapamycin (20 nM), where indicated, for 45 min prior to stimulation with LPS (20 µg/ml) and IL-4 (10 µg/ml) for 48 hr. (**A**) shows representative CTV staining, (**B**) percentage of B cells per generation quantified from (**A**), (**C**) live B cell number (**D**) percentage of live B cells (7AAD-ve), and (**E**) forward scatter. Data shows three technical replicates from cells isolated from one mouse and is representative of three independent experiments. Statistical power for (**B**) was determined using two-way ANOVA followed by multiple comparison testing via Sidak's analysis. For comparison to LPS + IL-4 stimulated B cells, $p < 0.0001$ by **** and ns by $>0.05$. (**F**) Splenocytes from C57BL6/J mice were pre-treated with DMSO as a vehicle control, PD18352 (2 µM), VX745 (1 µM), or Rapamycin (20 nM), where indicated, for 45 min prior to stimulation with LPS (20 µg/ml) and IL-4 (10 µg/ml). The cells were fixed after 24 hr and stained with filipin before acquisition. (**F**) Filipin staining after inhibitor treatment. Data shows three technical replicates from cells isolated from one mouse and is representative of three independent experiments. Statistical power for (**C-F**) was determined by one-way ANOVA followed by multiple comparison testing via Dunnett's analysis, where $p < 0.05$ is indicated by *, $p < 0.01$ by **, $p < 0.001$ by ***, $p < 0.0001$ by **** and ns by $>0.05$. For all panels, cells in the absence of LPS + IL-4 were naive B cells analysed on the day of isolation.

The online version of this article includes the following source data and figure supplement(s) for figure 9:

**Source data 1.** Raw FACS files for *Figure 9A–E* (proliferation – PD184352, VX745, Rapamycin), 9 F (filipin staining - PD184352, VX745, Rapamycin).

**Figure supplement 1.** MyD88 is required for signalling through TLR4.

**Figure supplement 1—source data 1.** PDF files containing labelled and uncropped images for western blots displayed in *Figure 9—figure supplement 1*.

**Figure supplement 1—source data 2.** Original files for western blot images displayed in *Figure 9—figure supplement 1*.

**Figure supplement 2.** Inhibition of mTOR impacts B cell function.

**Figure supplement 2—source data 1.** Raw FACS files for *Figure 9—figure supplement 2A* (OPP uptake – rapamycin), 2B (kynurenine uptake – rapamycin).

B cells treated with PD184352 remained in generation 1 with a smaller percentage in generation 4, compared to the LPS + IL-4 stimulation alone (*Figure 9B*). Treatment with VX745, a selective inhibitor of p38α and β (*McGuire et al., 2013*), had a lesser effect on proliferation, with the only difference being a lower percentage in generation 4 (*Figure 9A–B*). To determine the impact of inhibiting both p38/MEK1/2 signalling, we treated B cells with a combination of both PD184352 and VX745, which reduced proliferation to 3 generations of B cells with a higher proportion of B cells in generations 1 and 2 (*Figure 9A and B*). Inhibition of the mTOR pathway using rapamycin had the greatest effect on proliferation out of all the compounds tested, resulting in 70% of B cells remaining in generation

1, with few cells progressing past generation 2 (*Figure 9A–B*). This correlated with the number of live B cells, with combined MAPK inhibition and mTOR inhibition resulting in the biggest impact on B cell number (*Figure 9C*). Interestingly, most of the inhibitor treatments had minimal impact on the percentage of live cells at 48 hr, although treatment with PD184352 reduced the percentage of live cells by 7% and the combination of PD184352 and VX745 by 15% (*Figure 9D*). In terms of cell size, treatment with PD184352, the combination of PD184352 and VX745, and rapamycin reduced cell size (*Figure 9E*). Treatment with all the inhibitors reduced the levels of filipin staining, with combined PD184352 and VX745 treatment having the most significant effect on cholesterol levels (*Figure 9F*). Together, this would suggest multiple signalling pathways feed into the regulation of cholesterol in LPS-activated B cells. Interestingly, while the mTOR pathway is strongly linked to the regulation of protein synthesis (*Ma and Blenis, 2009*), rapamycin reduced but did not block protein synthesis and amino acid uptake in B cells stimulated with LPS +IL-4 (*Figure 9—figure supplement 2A, B*).

## The mevalonate pathway is required for B cell proliferation in response to multiple agonists

While the above results indicate that cholesterol is required for B cell proliferation in response to LPS + IL-4 stimulation, they do not demonstrate whether the primary signal for this is LPS or IL-4. To test this, B cells were stimulated with either IL-4, LPS, or a combination of both, and changes in cholesterol were measured by filipin staining. Stimulation with either LPS or IL-4 for 24 hr was able to increase cholesterol levels, although LPS had a greater effect (*Figure 10A*).

Considering the effects of HMGCR inhibition on cholesterol levels following LPS stimulation, we also tested whether other stimuli known to promote B cell proliferation could elevate cholesterol levels. Both the TLR7 agonist Resiquimod and the TLR9 agonist CpG increased cholesterol levels in B cells (*Figure 10B*). Similar to TLR agonists, both anti-IgM and CD40L stimulation increased cholesterol levels in B cells (*Figure 10B*). Likewise, Resiquimod, CpG, anti-IgM, and CD40L were all able to stimulate proliferation, as evidenced by CTV labelling and number of live B cells after 72 hr of treatment (*Figure 10D and E*). While none of these stimuli were as effective at inducing proliferation as LPS, in each case, treatment with Fluvastatin reduced survival and blocked proliferation of the B cells (*Figure 10C–E*).

Since stimulation with these agonists increased cholesterol levels in B cells, we wondered whether that correlated with an increased expression of key proteins involved in cholesterol metabolism at a proteomic level. Based on proteomic data from *James et al., 2026* stimulation of B cells with either IL-4, anti-IgM, or anti-CD40 could increase levels of HMGCR, SQLE, and LDLR, with the largest increase in expression seen with a combination of these stimuli (*Figure 10—figure supplement 1*).

## Discussion

In this study, we have used proteomic analysis to highlight the switch in metabolic demand from naive B cells in a resting state to LPS + IL-4 activated B cells, which significantly increase their energy and biosynthetic requirements to sustain growth, proliferation, and class switching.

Upon activation, B cells upregulated several amino acid transporters, including the system L transporters SLC7A5 and 6 (*Figure 4A*), which can import large neutral amino acids, including the essential amino acids histidine, isoleucine, leucine, methionine, phenylalanine, threonine, tryptophan, and valine (*Fotiadis et al., 2013*). Of these two transporters, SLC7A5 was expressed at a higher level and was responsible for the majority of amino acid uptake via system 1 transporters (*Figure 4J*). SLC7A5 forms a dimer with CD98 (SLC3A2) to form an active transporter. SLC7A5 is not, however, the only binding partner of CD98, as it can also bind to the SLC7A11 and SLC7A6 transporters or interact with integrins (*Johnstone et al., 2024*). Previous work has shown that mice with B cell-specific deletions of CD98 presented normal development of B cells in the bone marrow, but significantly lower levels of class-switched IgG due to the inhibition of B cell proliferation and plasma cell formation following the activation of naive B cells in the periphery (*Cantor et al., 2009*). The defect in proliferation was attributed to defective integrin signalling rather than an impairment in amino acid transport, based on the deletion of the integrin or amino acid transporter regions in CD98. Knockout of *Slc7a5* in either hematopoietic cells or B cells has been previously shown not to affect the number of splenic B cells in mice (*Sinclair et al., 2013*), although a separate study found a decrease in the number of peritoneal

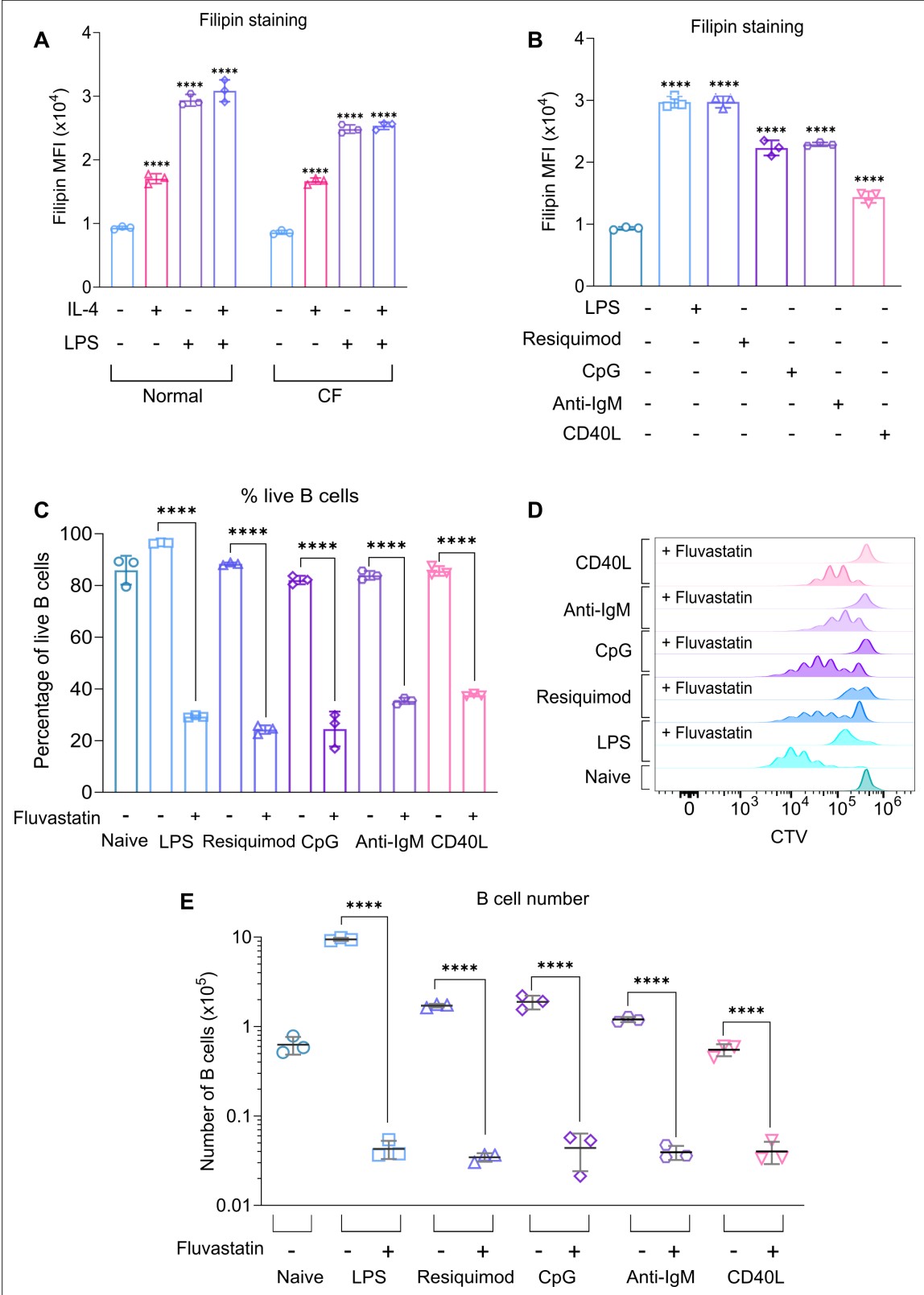

**Figure 10.** Cholesterol is required for the growth and proliferation of B cells by multiple stimuli. (**A**) Splenocytes from C57BL6/J mice were plated in normal or cholesterol-free (CF) media, before stimulation with LPS (20 µg/ml), IL-4 (10 µg/ml), or a combination of LPS and IL-4. The cells were fixed after 24 hr and stained with filipin. (**A**) Filipin staining comparing cholesterol content between different stimuli. Data shows the results of three biological replicates. Statistical power was determined using two-way ANOVA followed by multiple comparison testing via Tukey's analysis. For comparisons to

*Figure 10 continued on next page*

*Figure 10 continued*

naive B cells, p<0.0001 is indicated by ****. (**B**) Splenocytes from C57BL6/J mice were plated in normal media before stimulation with LPS (20 µg/ml), Resiquimod (1 µg/ml), ODN 1826 (1 µg/ml), anti-IgM (10 µg/ml), or CD40L (500 ng/ml). The cells were fixed after 24 hr and stained with filipin. (**B**) Filipin staining comparing cholesterol content between different stimuli. Data shows the results of three biological replicates. Statistical power was determined using one-way ANOVA followed by multiple comparison testing via Dunnett's analysis. For comparisons to naive B cells, p<0.0001 is indicated by ****. (**C–E**) B cells were purified from the spleens of C57BL6/J mice and cultured in normal media. The cells were stained with CTV, then pre-treated with DMSO as a vehicle control or Fluvastatin (10 µM) for 45 min before stimulation with LPS (20 µg/ml), Resiquimod (1 µg/ml), ODN 1826 (1 µg/ml), anti-IgM (10 µg/ml), or CD40L (500 ng/ml) as indicated for 72 hr. (**C**) shows the percentage of live B cells (7AAD-ve), (**D**) representative histogram for CTV staining, and (**E**) live B cell number. Data shows the results of three biological replicates. Statistical power was determined using two-way ANOVA followed by multiple comparison testing via Tukey's analysis, where p<0.0001 is indicated by ****. For (**E**), this data was log-transformed and then statistically analysed due to unequal variance. Statistical power was determined using two-way ANOVA followed by multiple comparison testing via Sidak's analysis, where p<0.0001 is indicated by ****.

The online version of this article includes the following source data and figure supplement(s) for figure 10:

**Source data 1.** Raw FACS files for *Figure 10A* (filipin staining – IL-4, LPS, LPS +IL-4 [normal/CF]), 10B (filipin staining – all agonists).

**Source data 2.** Raw FACS files for *Figure 10C–D* (proliferation – all agonists).

**Figure supplement 1.** IL-4, anti-IgM, and anti-CD40 stimulation upregulate proteins involved in cholesterol metabolism in B cells.

B1 cells (*Sun et al., 2023*). While not critical for initial B cell development, SLC7A5 was critical for the survival and expansion of B cells in response to ex vivo stimulation with LPS + IL-4 (*Figure 4F–I*). Of note, recent research in human B cells has shown that stimulation with CpG increases SLC7A5 expression, which in turn transports L-leucine into the cell and facilitates mTORC activation, which was required for IgG production and inflammatory cytokine secretion (*Torigoe et al., 2019*).

Similar observations have been made in T cells, where SLC7A5 is not required for initial T cell development but is required for the clonal expansion and effector functions of T cells following activation via the T cell receptor in the periphery (*Sinclair et al., 2013*). This suggests that B cells, similar to T cells, may rely on different amino acid transporters at different points in their development. The expression of SLC7A5 following T cell activation has been shown to require the transcription factor c-Myc (*Marchingo et al., 2020*). While the role of c-Myc was not directly addressed in this study, analysis of the proteomic data presented here does show that c-Myc was increased by LPS stimulation (*Supplementary file 3*). This is consistent with previous reports showing an upregulation of c-Myc in response to several B cell activating stimuli, including LPS, CD40, and the BCR (*Luo et al., 2018*; *Dominguez-Sola et al., 2012*; *Calado et al., 2012*; *Tesi et al., 2019*). While in T cells, c-Myc regulates metabolic changes, including the expression of SLC7A5 (*Sinclair and Cantrell, 2025*), this has not been studied to the same degree in B cells. Of note, transcriptional profiling following LPS stimulation of B cells has suggested that *Slc7a5* mRNA induction in response to LPS can occur in the absence of c-Myc (*Tesi et al., 2019*), suggesting that the regulation of SLC7A5 may not be identical between T and B cells.

As B cells expand and divide, they will require an increase in lipids to generate new membrane. Cholesterol is an important lipid component of the plasma membrane, which affects membrane fluidity (*Luo et al., 2020*) and has also been implicated in the regulation of several receptors, including the BCR (*Bléry et al., 2006*). The regulation of cholesterol levels in cells is therefore closely controlled at the level of both cholesterol biosynthesis and its uptake via the endocytosis of LDL particles. Low levels of cholesterol in the endoplasmic reticulum are detected by the transcription factor SREBP2 (*Luo et al., 2020*; *Brown et al., 2018*). The importance of this system in B cells was recently demonstrated by the conditional knockout of SCAP in B cells, a key regulator of SREBP1 and 2. While this did not prevent the development of B cells or their levels at steady state in mice, it did prevent the activation and proliferation of B cells in response to antigens and thus compromised the humoral immune response (*Luo et al., 2023*). SCAP deletion will, however, affect both SREBP1 and SREBP2 function and therefore have wider effects on lipid metabolism beyond its effects on cholesterol. We show here that in response to several activating stimuli ex vivo, including LPS, B cells upregulate cholesterol, and that blocking HMGCR, a key rate-limiting enzyme in the cholesterol biosynthesis pathway, inhibits B cell proliferation (*Figure 10B–E*). In the case of LPS stimulation, B cells seem to be able to obtain cholesterol by both uptake and biosynthesis. For example, B cells can still increase their cholesterol levels in the absence of LDL in the cell culture medium (*Figure 5I*). In addition, NB-598, an inhibitor of the 2nd rate-limiting enzyme in the cholesterol biosynthesis

pathway, can reduce the ability of LPS to increase cholesterol levels (*Figure 5I*). In contrast, the greater effect of NB-598 on cholesterol levels of B cells grown in CF media relative to normal media (*Figure 5I*), and the inability of mevalonate to rescue the effects of statin treatment in CF media (*Figure 7B–F*), would indicate that B cells also use LDL uptake as a source of cholesterol. In line with this, B cells upregulated the LDLR following activation (*Figure 5D*). Additionally, proteomic analysis of LPS + IL-4 stimulated B cells grown in normal vs CF media, or +/-Fluvastatin treatment reveals an upregulation of proteins involved in cholesterol metabolism, including HMGCR, SQLE and LDLR (*Figure 6—figure supplement 1*). This suggests that a feedback mechanism exists whereby the detection of low cholesterol levels (either due to blocking biosynthesis or uptake) by SREBP2 increases the expression of cholesterol metabolic proteins. A similar study in activated CD8+T cells has demonstrated that knockout of the LDLR leads to an upregulation of proteins involved in cholesterol biosynthesis, including HMGCR, and a reduction in proteins involved in cholesterol efflux compared to WT CD8+T cells (*Bonacina et al., 2022*). The precise balance between these two mechanisms in B cells may depend on the duration of stimulation and the availability of LDL in the extracellular environment.

In line with our results, following CD40 mediated activation of human B cells, Atorvastatin was reported to inhibit human B cell proliferation and the expression of the activation markers CD80 and CD86 (*Shimabukuro-Vornhagen et al., 2014*). Interpretation of the effects of statins on B cells is, however, complex. Statins inhibit HMGCR, the rate-limiting enzyme in the mevalonate pathway that is required for the initial steps in cholesterol biosynthesis. This pathway generates farnesyl pyrophosphate, an intermediate that can feed into not only cholesterol biosynthesis but also protein prenylation and ubiquinone pathways (*Mullen et al., 2016*; *Wang and Casey, 2016*). Thus, the effects of statins reported here could be due not only to their effects on cholesterol but also via these other pathways. Prenylation has been linked to proliferation in several cell types (*Su et al., 2020*), including T cells, where Simvastatin inhibits activation via two mechanisms, one of which involves blocking the prenylation of Ras and Rac (*Ghittoni et al., 2005*). We therefore investigated whether statin-mediated effects on B cell survival and proliferation were in part due to prenylation. We found that FGTI-2734, a dual inhibitor of FTase and GGTase, blocked LPS + IL-4 induced B cell proliferation and survival (*Figure 6E–H*). Similarly GGTI-298, an inhibitor of GGTase, had a stronger inhibitory effect on B cell function compared to FTI-277, which blocks FTase activity, suggesting an important role for prenylation in B cells (*Figure 6I–P*). In agreement with this, prenylation has also been reported to be important in B cells for the expression of CD80 and CD86 following stimulation via CD40, although its role in proliferation was not directly addressed (*Shimabukuro-Vornhagen et al., 2014*). Given that prenylation can target a large number of proteins, including many members of the Ras GTPase superfamily, to regulate their cellular localisation, the effects of blocking prenylation in B cells are likely to be complex. Interestingly, GGPP, which is needed for protein geranylgeranylation, was able to rescue the effects of Fluvastatin on LPS + IL-4 stimulated B cells in normal media (*Figure 8A–D*). Notably, it was more efficient than mevalonate at restoring proliferation in the Fluvastatin treated B cells. This ability to rescue proliferation was unexpected, as GGPP would not directly feed into the cholesterol biosynthesis pathway, and other experiments described here using CF media and squalene monooxygenase inhibitors argue for a requirement of cholesterol for B cell proliferation. We therefore looked at its effect on cholesterol levels and found that GGPP was able to rescue the effects of Fluvastatin in normal but not CF media (*Figure 8I and J*). This could be explained by prenylation being required for cholesterol uptake but not biosynthesis. Cholesterol uptake via the LDLR requires the clathrin-mediated endocytosis of the LDLR and its trafficking to endosomes, where cholesterol is released from LDL (*Nguyen et al., 2025*). Vesicle trafficking is regulated via the action of several Rab GTPases, including Rab5, which is recruited to early endosomes and regulates their maturation (*Zhao et al., 2025*; *Zeigerer et al., 2012*). More significantly, Rab5 prenylation has been shown to be important for its correct targeting to endosomes, and therefore its biological function (*Gomes et al., 2003*). Treatment of cells with statins to inhibit prenylation has previously been shown to affect Rab subcellular localisation and vesicle trafficking (*Ali et al., 2010*; *Ronzier et al., 2019*; *Xia et al., 2018*). Loss of the prenylation of Rab proteins involved in LDLR endocytosis could therefore negatively impact this process, leading to a failure of LDL uptake into the B cell. Prenylation is also likely to impact other processes in B cells, for example, TLR9-induced IL-10 production by regulatory B cells has been found to require geranylgeranylation and not farnesylation (*Bibby et al., 2020*). Further work will be

necessary to identify the proteins prenylated upon B cell activation and to determine the effects of prenylation on their function.

In terms of the pathways that control B cell function downstream of TLR4 signalling, we found that inhibition of p38α/β and MEK1/2 signalling impacted B cell growth and proliferation, as well as cholesterol levels (*Figure 9A–F*). Related to this, inhibition of MEK1/2 has been found to abolish *LDLR* promoter activity (*Kotzka et al., 2000*), and depletion of ERK2, but not ERK1, significantly reduced the expression levels of *SREBP2, HMGCR,* and *LDLR* (*Yu et al., 2019*). Studies in primary mouse hepatocytes have demonstrated that treatment with the p38α/β inhibitor SB203580 also significantly reduced *LDLR* promoter activity (*Pham et al., 2016*).

Additionally, we found that blocking mTOR had a major inhibitory effect on B cell proliferation (*Figure 9A–B*), as well as reducing cholesterol levels (*Figure 9F*), and to a lesser extent, the rate of protein synthesis and amino acid uptake through System L transporters (*Figure 9—figure supplement 2*). In response to environmental cues, including energy and nutrient availability, the mTORC1 complex regulates processes such as cell growth and proliferation (*Saxton and Sabatini, 2017*). Previous studies have shown that the addition of amino acids to the growth medium increases the transcription of genes involved in cholesterol biosynthesis, including HMGCR and SREBP2 (*Eid et al., 2017*). Treatment with the mTORC1 inhibitor Torin1 blocked this effect, suggesting that cholesterol metabolism may be dependent on mTORC1 activity (*Eid et al., 2017*).

Overall, the proteomic analysis described here supports that B cells undergo a major reorganisation of metabolic pathways following TLR activation. Naive B cells exist in a quiescent state where they do not grow or proliferate and express low levels of metabolic enzymes. Conversely, B cells stimulated with LPS + IL-4 massively upregulate metabolic enzymes involved in processes including protein synthesis, amino acid uptake, and cholesterol metabolism, which is required to support an increase in cell size, proliferation, and antibody class switching. Most significantly, we found that LPS + IL-4 stimulated B cells depended on cholesterol biosynthesis, cholesterol uptake, and prenylation for expansion and proliferation.

## Methods
### Mice
Wildtype C57BL6/J mice (Charles Rivers) aged between 8 and 12 weeks were used for the majority of experiments. Conditional SLC7A5 knockout mice bred to Vav-iCre mice (*Slc7a5$^{fl/fl}$/Tg(Vav1-iCre)$^{+/-}$*) have been described previously and were maintained on a C57BL6/J background (*de Boer et al., 2003*; *Poncet et al., 2014*; *Sinclair et al., 2013*). MyD88$^{-/-}$ (MyD88 KO) mice have been described previously and were maintained on a C57BL6/J background (*Adachi et al., 1998*). The mice were kept in individually ventilated cages under specific pathogen-free conditions. Mice were given free access to food (RM3 irradiated pelleted diet, Special Diet Services) and water. Animal rooms were maintained at 21 °C and a humidity of 45–55% with a 12/12 hr light cycle. Mice were sacrificed using a rising concentration of $CO_2$, and death was confirmed by either cervical dislocation or exsanguination. This work was performed under a U.K. Home Office Project Licence (70/7901) in compliance with UK Home Office Animals (Scientific Procedures) Act 1986 guidelines, and approved by the University of Dundee's Welfare and Ethical Use of Animals Committee.

### Cell isolation
Lymph nodes were homogenised through a 40 μM cell strainer in DPBS buffer (14190144, Gibco), and after centrifugation at 450 × *g* for 4 min, the pellet was resuspended in DPBS. Lymphocytes were counted on a Novocyte (Agilent Technologies) and seeded at 1×10$^6$ cells/ml in B cell medium (RPMI-1640 supplemented with 10% FBS, 50 U/ml penicillin-streptomycin, 5 mM L-glutamine, 10 mM HEPES buffer, 1 mM sodium pyruvate, 50 μM 2-mercaptoethanol and non-essential amino acids) for stimulation.

Spleens were dissociated through a 40 μM filter in DPBS buffer, and after centrifugation at 450 × *g* for 4 min, the pellet was resuspended in 1 ml of Red Blood Cell Lysing Buffer Hybri-Max (R7757, Sigma) for 4 min at room temperature to lyse RBCs. The reaction was quenched by the addition of DPBS, and the cells were pelleted by centrifugation at 450 × *g* for 4 min. The pellet was resuspended in 10 ml

**Table 1.** MACS antibodies.

| Antibody | Clone | Concentration | Catalogue number | Supplier |
|---|---|---|---|---|
| Anti-CD11b | M1/70 | 10 µg/ml | 101204 | BioLegend |
| Anti-CD11c | N418 | 10 µg/ml | 117304 | BioLegend |
| Anti-CD3ε | 145–2 C11 | 10 µg/ml | 100304 | BioLegend |
| Anti-TER119 | TER-119 | 10 µg/ml | 116204 | BioLegend |

DPBS before counting the splenocytes on a Novocyte. Splenocytes were seeded at $1\times10^6$ cells/ml in B cell medium for stimulation or used for further purification of B cells.

## B cell isolation

To obtain an enriched population of B cells, magnetic sorting was used to remove non-B cells. Briefly, splenocytes were incubated with Fc receptor-blocking anti-CD16/CD32 antibody (553141, BD Pharmingen) for 5 min on ice. The cells were incubated with a mixture of biotin-labelled anti-mouse antibodies (10 µg/ml; *Table 1*) for 20 min on ice. Unbound antibody was removed by washing with 10 ml Magnetic Activated Cell Sorting (MACS) buffer (DPBS +0.5% BSA+2 mM EDTA), followed by centrifugation. The cells were then resuspended in MACS buffer with streptavidin microbeads (1:10; 130-048-101, Miltenyi Biotech) and incubated at 4 °C for 15 min. Miltenyi LD columns (130-042-901, Miltenyi Biotech) were calibrated by passing through 3 ml of MACS buffer. Cells were washed with 10 ml MACS buffer and centrifuged at 450 × *g* for 4 min to remove excess beads. The cell pellet was then resuspended in MACS buffer and separated using magnetic-activated cell sorting columns (LD) according to the manufacturer's instructions (Miltenyi Biotech). Single-cell suspensions were collected, and B cell numbers and purity (CD19+) were measured on a NovoCyte. Typical purity levels were 97% CD19+B cells. B cells were seeded at $2\times10^5$ cells/ml of B cell medium for stimulation for flow cytometry experiments unless otherwise stated. When stated, purified B cells were stained with 2.5 µM CellTrace Violet (C34557, Thermo Fisher) at 37 °C for 20 min, then quenched in 10 ml B cell medium before seeding at the required density.

## Stimulations and inhibitors

Cells were rested at 37 °C for 1 hr. When required, the cells were pre-treated with the appropriate inhibitor (*Table 2*) for 45 min before stimulation, at the concentrations indicated in the figure legend. Inhibitors were dissolved in DMSO (D2650, Sigma Aldrich) and used at a dilution of 1:1000 of culture medium. As indicated in the figure legends, cells were treated with 1 µl DMSO as a vehicle control.

**Table 2.** Inhibitors.

| Name | Target | Concentration | Catalogue number | Supplier |
|---|---|---|---|---|
| 2-Aminobicyclo-(2,2,1)-heptane-2-carboxylic acid (BCH) | LAT1 | 10 mM | A7902-1G | Sigma |
| Fluvastatin sodium | HMG-CoA reductase | 0.1 µM –100 µM | 3309 | Tocris |
| Rosuvastatin calcium | HMG-CoA reductase | 0.1 µM –100 µM | 6343 | Tocris |
| NB-598 | Squalene monooxygenase | 0.1 µM –100 µM | HY-16343 | MedChem Express |
| FGTI-2734 | Farnesyl and geranylgeranyl transferase-1 | 10 µM | HY-128350 | MedChem Express |
| FTI-277 hydrochloride | Farnesyl transferase | 10 µM-30µM | HY-15872A | MedChem Express |
| GGTI-289 trifluoroacetate | Geranylgeranyl transferase-1 | 10 µM-30µM | HY-15871 | MedChem Express |
| PD18352 | MSK1/2 | 2 µM | 4237 | Tocris |
| VX745 | P38 | 1 µM | 3915 | Tocris |
| Rapamycin | mTOR | 20 nM | R8781 | Merck |

**Table 3.** Flow cytometry antibodies.

| Antibody | Fluorophore | Clone | Dilution | Catalogue number | Supplier |
|---|---|---|---|---|---|
| Anti-CD19 | APC | 6D5 | 1:300 | 115512 | Biolegend |
| Anti-CD19 | PE | 1D3 | 1:200 | 557399 | BD Biosciences |
| Anti-CD19 | FITC | 1D3 | 1:200 | 152403 | BioLegend |
| Anti-CD93 | APC | AA4.1 | 1:200 | 136510 | BioLegend |
| Anti-CD98 | PE | RL388 | 1:200 | 128208 | BioLegend |
| dsDNA | eFluor 660 | - | 1:1000 | 65-0864-14 | eBioscience |
| Anti-IgG1 | FITC | A85-1 | 1:200 | 553443 | BD Biosciences |
| Anti-pRb | FITC | D20B12 | 1:300 | 4277 S | Cell Signaling Technology |

Cells were pre-incubated with the following metabolites at 37 °C for 1 hr before inhibitor treatment/stimulation. Metabolites were used at the concentrations indicated in the figure legend: Mevalonic acid 5-phosphate lithium salt hydrate (79849, Sigma Aldrich), which was dissolved in HEPES (15630080, Thermo Fisher). As indicated in the figure legends, cells were treated with 20 µl HEPES as a vehicle control. Geranylgeranyl pyrophosphate ammonium salt (G6025, Sigma Aldrich), which comes in a 7:3 methanol: ammonium hydroxide solution. As indicated in the figure legends, cells were treated with 4.55 µl 7:3 methanol: ammonium hydroxide solution as a vehicle control.

A combination of 20 µg/ml LPS (L2654, Sigma) and 10 ng/ml IL-4 (214–14, Peprotech) was used to stimulate the cells unless otherwise stated in the figure legend. For some experiments, cells were treated with LPS or IL-4 alone, or with IFN-β (10 ng/ml; 581302, Biolegend), 1 µg/ml Resiquimod/R848 (tlrl-r848, Invivogen), 1 µg/ml ODN 2006 (tlrl-2006, Invivogen), 10 µg/ml Anti-Mouse IgM µ chain (115-006-075, Stratech), or 500 ng/ml CD40 ligand (8230 CL-050/CF, R&D).

## Cholesterol-free serum

Silica-depleted FBS was purchased from Biowest (S140L) and used in place of normal FBS when making B cell culture medium. Samples of normal and cholesterol-free FBS were sent to London Health Company, which carried out tests to measure changes in triglycerides, cholesterol, high-density lipoprotein (HDL), and low-density lipoprotein (LDL) (*Supplementary file 2*).

## Cell survival, proliferation, and class switching assays

Following stimulation, cells were washed in Fluorescence Activated Cell Sorting (FACS) buffer (1% BSA, DPBS) and then incubated with Fc receptor-blocking anti-CD16/CD32 antibody in FACS buffer for 15 min at 4 °C before staining with the appropriate fluorochrome-conjugated antibodies (*Table 3*) in FACS buffer for 20 min at 4 °C. The samples were washed with FACS buffer prior to resuspension in 0.25 µg of 7AAD viability staining solution (15239004, Invitrogen). Flow cytometry data was collected on a Novocyte and analysed using FlowJo V.10 software (BD Biosciences). A representative gating strategy is shown in *Figure 1—figure supplement 4A* for lymph node/splenocyte isolation, and *Figure 4—figure supplement 3C* for B cell purification.

## Mitochondria staining assay

Splenocytes were stimulated with 20 µg/ml LPS and 10 ng/ml IL-4. After 24 hr, the cells were washed twice in FACS buffer. 50 µl of prewarmed (37 °C) staining solution containing 100 nM MitoTracker Red FM probe (M22425, Invitrogen) in FACS buffer was added to the cells before incubation at 37 °C for 20 min. The samples were washed with FACS buffer before incubation with Fc receptor-blocking anti-CD16/CD32 antibody in FACS buffer for 15 min at 4 °C before staining with the appropriate fluorochrome-conjugated antibodies (*Table 3*) in FACS buffer for 20 min at 4 °C. The samples were washed with FACS buffer prior to resuspension in 0.25 µg of 7AAD. Flow cytometry data was collected on a Novocyte and analysed using FlowJo V.10 software. A representative gating strategy is shown in *Figure 1—figure supplement 4B*.

**Table 4.** Western blot primary antibodies.

| Antibody target | Animal raised in | Dilution | Code | Source |
|---|---|---|---|---|
| Phospho-STAT1 (Tyr701) | Rabbit | 1:1000 | 9167 | Cell Signalling Technology |
| Phospho-p44/42 MAPK (Erk1/2) (Thr202/Tyr204) | Rabbit | 1:1000 | 9101 | Cell Signalling Technology |
| p44/42 MAPK (Erk1/2) | Rabbit | 1:1000 | 9102 | Cell Signalling Technology |
| Phospho-p38 MAPK (Thr180/Tyr182) | Rabbit | 1:1000 | 4511 | Cell Signalling Technology |
| p38 MAPK | Rabbit | 1:1000 | 9212 | Cell Signalling Technology |
| Phospho-Akt (Ser473) | Rabbit | 1:1000 | 4060 | Cell Signalling Technology |
| Phospho-p70 S6 Kinase (Thr389) | Rabbit | 1:1000 | 9234 | Cell Signalling Technology |
| Phospho-S6 Ribosomal Protein (Ser235/236) | Rabbit | 1:1000 | 2211 | Cell Signalling Technology |
| S6 Ribosomal Protein (5G10) | Rabbit | 1:1000 | 2217 | Cell Signalling Technology |

## p-STAT1 staining

Splenocytes ($1\times10^6$ cells/ml) were stimulated with LPS (20 µg/ml) and/or IFN-β (10 ng/ml) for 15 min before washing in FACS buffer. The cells were fixed using 1:1 IC fixation buffer (catalogue number: 00-8222-49, eBiosciences) in FACS buffer and incubated at room temperature for 20 min. Following another wash in FACS buffer, cells were resuspended in 100 µl of cold ethanol and stored at –20 °C for 30 min. The samples were washed with FACS buffer twice before incubation with Fc receptor-blocking anti-CD16/CD32 antibody in FACS buffer for 15 min at 4 °C. The cells were incubated with p-STAT1 (*Table 4*) for 1 hr at room temperature before washing with FACS buffer and incubation with (1:1000) anti-rabbit IgG (H+L), F(ab')$_2$ Fragment (Alexa Fluor 647 Conjugate; Cell Signalling Technology, 4414 S) and CD19 PE (*Table 3*) for 1 hr at room temperature. The cells were washed with FACS buffer before resuspension in 300 µl FACS buffer. Flow cytometry data was collected on a Novocyte and analysed using FlowJo V.10 software.

## Cell cycle analysis and p-Rb staining

Cells were washed in FACS buffer and then fixed using 1:1 IC fixation buffer and FACS buffer, and incubated at room temperature for 20 min. Following another wash in FACS buffer, cells were resuspended in 100 µl of cold ethanol and stored at –20 °C for 30 min. The samples were washed with FACS buffer twice before incubation with Fc receptor-blocking anti-CD16/CD32 antibody in FACS buffer for 15 min at 4 °C. Surface stains (*Table 3*) were incubated for 1 hr at room temperature before washing with FACS buffer and resuspension in 0.5 µg/ml of DAPI. Flow cytometry data was collected on a Novocyte and analysed using FlowJo V.10 software. A representative gating strategy is shown in *Figure 1—figure supplement 4C*.

## O-propargyl-puromycin uptake assay

Uptake of the puromycin analogue O-propargyl-puromycin was used to estimate protein synthesis rates. Splenocytes ($1\times10^5$ cells/100 µl) were stimulated with 20 µg/ml LPS and 10 ng/ml IL-4. After 24 hr, the cells were treated with 20 µM of O-propargyl-puromycin (NU-931–5, Jena Bioscience) for 30 min at 37 °C. 100 µl of 2% paraformaldehyde in B cell medium was added to the cells before incubation in complete darkness at room temperature for 30 min. The samples were pelleted by centrifugation at $450 \times g$ for 4 min before resuspension in 100 µl of DPBS. Following this, the samples were washed with FACS buffer and then resuspended in 100 µl of 0.01% Saponin (SAE0073, Sigma) in DPBS, before incubation at room temperature for 20 min. The samples were pelleted by centrifugation

at 450 × $g$ for 4 min before resuspension in 30 µl of Click mixture: 1 mM Copper sulphate (209198, Sigma), 10 mM Sodium ascorbate (A7631. Sigma), 1 mM BTTAA (906328, Sigma), 10 mM Amino-guanidine (81530, Cayman Chemicals), DPBS (14190144, Gibco), 5 µM Alexafluor 647 azide (A10277, Invitrogen) and incubation in complete darkness at room temperature for 1 hr. The samples were pelleted by centrifugation at 450 × $g$ for 4 min before resuspension in 200 µl of FACS buffer and incubation in complete darkness at room temperature for 30 min. Again, the samples were pelleted by centrifugation at 450 × $g$ for 4 min before washing again in FACS buffer. The cells were incubated with Fc receptor-blocking anti-CD16/CD32 antibody in FACS buffer for 15 min at 4 °C. Surface stains (*Table 3*) were incubated for 1 hr at room temperature. The samples were washed with FACS buffer prior to resuspension in 300 µl FACS buffer. Flow cytometry data was collected on a Novocyte and analysed using FlowJo V.10 software. A representative gating strategy is shown in *Figure 1—figure supplement 4D*.

## Kynurenine uptake assay

Uptake of the tryptophan metabolite L-kynurenine (K8625, Sigma) was used to estimate uptake rates via neutral amino acid transporters (*Sinclair et al., 2018*). Purified B cells (1×10⁶ cells/ml) were stimulated with 20 µg/ml LPS and 10 ng/ml IL-4. For the assay, pre-warm HBSS, 2-aminobicyclo-(2,2,1)-h eptane-2-carboxylic acid (BCH; 40 mM in HBSS; *Table 2*) and Kynurenine (800 µM in HBSS) to 37 °C. After 24 hr, the cells were washed in FACS buffer and then incubated with Fc receptor-blocking anti-CD16/CD32 antibody in FACS buffer for 15 min at 37 °C before staining with CD19 APC (*Table 3*) in FACS buffer for 20 min at 37 °C. The samples were washed with FACS buffer and then resuspended in 200 µl HBSS. Keep the samples at 37 °C. Either 100 µl of HBSS or 100 µl of BCH was then added to each sample. After this, 100 µl of Kynurenine was added, and the samples were incubated for 5 min at 37 °C. After 5 min, the samples were fixed by adding 125 µl of 4% paraformaldehyde made in DPBS for 30 min at room temperature. The samples were then washed two times with FACS buffer and re-suspended in 400 µl of FACS buffer. Flow cytometry data was collected on an LSR Fortessa (BD biosciences) and analysed using FlowJo V.10 software. A representative gating strategy is shown in *Figure 4—figure supplement 3A*.

## Cholesterol staining assay

Changes in cellular cholesterol content were determined using the cholesterol-binding agent filipin (F9765, Sigma; *Miller, 1984*). Splenocytes were stimulated with 20 µg/ml LPS and 10 ng/ml IL-4. After 24 hr, the cells were washed twice in DPBS before staining for viability using eFluor 660 fixable viability dye (*Table 3*) for 30 min at 4 °C. After two washes in FACS buffer, the cells were fixed using 1:1 IC fixation buffer and FACS buffer, before incubation at 4 °C for 20 min. Following another wash in FACS buffer, cells were incubated with Fc receptor-blocking anti-CD16/CD32 antibody in FACS buffer for 15 min at 4 °C. Surface stains (*Table 3*) were incubated for 20 min at 4 °C. The samples were washed with FACS buffer prior to resuspension in 270 µl FACS buffer and 30 µl of 1 µg/µl filipin (Merck) before incubation in complete darkness at room temperature for 30 min. Flow cytometry data was collected on an LSR Fortessa and analysed using FlowJo V.10 software. A representative gating strategy is shown in *Figure 4—figure supplement 3B*.

## Immunoblotting

B cells were seeded at 2x10⁶ cells/ml of B cell medium for immunoblotting. To generate lysates, the cells were stimulated with LPS (20 µg/ml) for 30 min and centrifuged at 100 × $g$ for 2 min. The pellet was resuspended in 300 µl of SDS Triton Lysis buffer (50 mM Tris-HCl pH 7.5, 1% (v/v) Triton-X-100, 1 mM EGTA, 1 mM EDTA, 1 mM Sodium Orthovanadate, 50 mM Sodium fluoride, 1 mM Sodium Pyro-phosphate, 10 mM Sodium B-glycerophosphate, 0.27 M Sucrose, 0.1% (v/v) 2-β-mercaptoethanol, cOmplete mini EDTA-Free Protease Inhibitor, 1% Sodium Dodecyl sulfate, 5% Glycerol) and incubated for 5 min at 100 °C to denature proteins. Following a brief period of cooling, the DNA was sheared using a 25 g syringe and the samples were stored at –20 °C. Cell lysates were resolved by SDS-PAGE using 10% gels in running buffer (250 mM Tris, 1% (w/v) 10% Sodium Dodecyl Sulfate, 192 mM glycine) and transferred to nitrocellulose membranes in the presence of transfer buffer (48 mM Tris, 39 mM glycine, 20% (v/v) methanol). The membranes were blocked using 5% (w/v) skimmed milk powder (Marvel) in TBST (0.5 M Tris-HCl, pH 7.6, 1.5 M NaCl and 0.1% (v/v) Tween 20). Primary antibodies

(*Table 4*) were used at a dilution of 1/1000 overnight in TBST with 5% (w/v) BSA. The membranes were incubated with Anti-rabbit HRP (horseradish peroxidase) secondary antibodies (Sigma) in 5% (w/v) milk powder in TBST. The blots were developed with Clarity Western ECL substrate (Bio-Rad) and scanned using Li-Cor Odyssey Fc Imager (Licor; Chemi, 600 nm and 700 nm channels). Image Studio software was utilised to compile western blot data.

## Proteomics

For the naive vs LPS + IL-4 dataset, wildtype C57BL/6 mice (Charles Rivers) aged between 8 and 12 weeks were used for all proteomic experiments. Lymph nodes were extracted from mice, mashed in RPMI media before filtering through a 70 µm cell strainer. Naive B cells were purified by fluorescence-activated cell sorting. Lymphocytes were incubated with Fc receptor-blocking anti-CD16/CD32 antibody in FACS buffer for 15 min at 4 °C before staining with fluorochrome-conjugated antibodies (*Table 3*) in FACS buffer for 20 min at 4 °C. The samples were washed with FACS buffer prior to resuspension in 0.5 µg/ml of DAPI. CD19+, CD93-, DAPI- cells were sorted using a Sony SH800 cell sorter. Sorted cells were washed twice with HBSS and snap frozen in liquid nitrogen to be stored at –80 °C until processing for mass spectrometry. For B cell activation, lymphocytes were suspended at a final density of 1.5 million cells/ml in medium (RPMI-1640 containing glutamine supplemented with 10% FBS, 50 U/ml penicillin-streptomycin, 50 µM 2-mercaptoethanol) and activated for 24 hr in the presence of 20 µg/ml LPS and 10 ng/ml IL-4. After 24 hr, the cells were incubated with Fc receptor-blocking anti-CD16/CD32 antibody in FACS buffer for 15 min at 4 °C before staining with fluorochrome-conjugated antibodies (*Table 3*) in FACS buffer for 20 min at 4 °C. The samples were washed with FACS buffer prior to resuspension in 0.5 µg/ml of DAPI. CD19+and DAPI- were sorted and collected as described above. Cells were then washed three times in PBS and cell pellets were frozen at –80 °C.

For the dataset in *Figure 6—figure supplement 1*, B cells were isolated from murine spleens as described above and seeded at $2x10^5$ cells/ml of B cell medium, with $1×10^6$ cells/5 ml per condition. Cells were rested at 37 °C for 1 hr. Cells were washed twice with PBS and snap frozen in liquid nitrogen to be stored at –80 °C until processing for mass spectrometry. For activation, B cells were pre-treated with Fluvastatin (10 µM) for 45 min before stimulation. The cells were stimulated with LPS (20 µg/ml) and IL-4 (10 ng/ml) for 24 hr. Cells were washed twice with PBS and snap frozen in liquid nitrogen to be stored at –80 °C until processing for mass spectrometry.

Cell pellets were lysed, and proteins were digested following the protocol described by *Baker et al., 2022*. In brief, 400 ml of lysis buffer (5% sodium dodecyl sulfate, 50 mM triethylammonium bicarbonate (pH 8.5) and 10 mM tris(2-carboxyethyl) phosphine-hydrochloride) was added to each sample and the lysates were shaken at room temperature at 1000 rpm for 5 min before being incubated at 95 °C at 500 rpm for 5 min. Samples were allowed to cool and were then sonicated using a BioRuptor (15 cycles: 30 s on and 30 s off) and alkylated with 20 mM iodoacetamide for 1 hr at 22 °C in the dark. To determine protein concentration, the EZQ protein quantitation kit (Thermo) was used, and protein cleanup and digestion were performed using S-TRAP mini columns (Protifi). Proteins were digested with trypsin at a 1:20 ratio (enzyme:protein) for 2 hr at 47 °C. Digested peptides were eluted from S-TRAP columns using 50 mM ammonium bicarbonate, followed by 0.2% aqueous formic acid and 50% aqueous acetonitrile containing 0.2% formic acid. Peptides were dried by speedvac before resuspending in 1% formic acid. Peptide quantity was measured using the CBQCA kit (Thermo Fisher).

## LC-MS/MS analysis

### Naïve vs LPS + IL-4 dataset

Peptides were analysed by single-shot Data Independent Acquisition (DIA) mass spectrometry as previously described (*Molina-Gonzalez et al., 2023*; *Sollberger et al., 2024*). 1.5 µg of peptide from each sample was injected onto a nanoscale C18 reverse-phase chromatography system (UltiMate 3000 RSLC nano, Thermo Scientific) and electrosprayed into an Orbitrap Exploris 480 Mass Spectrometer (Thermo Fisher). The following buffers were used for liquid chromatography: buffer A (0.1% formic acid in Milli-Q water (v/v)) and buffer B (80% acetonitrile and 0.1% formic acid in Milli-Q water (v/v)). Samples were loaded at 10 µl/min onto a trap column (100 µm × 2 cm, PepMap nanoViper C18 column, 5 µm, 100 Å, Thermo Fisher Scientific) equilibrated in 0.1% trifluoroacetic acid (TFA). The trap column was washed for 3 min at the same flow rate with 0.1% TFA, then switched in-line with a Thermo Fisher Scientific, resolving C18 column (75 µm × 50 cm, PepMap RSLC C18 column, 2 µm, 100 Å). Peptides were eluted from the

column at a constant flow rate of 300 nl/min with a linear gradient from 3% buffer B to 6% buffer B in 5 min, then from 6% buffer B to 35% buffer B in 115 min, and finally to 80% buffer B within 7 min. The column was then washed with 80% buffer B for 4 min and re-equilibrated in 3% buffer B for 15 min. Two blanks were run between each sample to reduce carryover. The column was kept at a constant temperature of 50 °C.

The data was acquired using an easy spray source operated in positive mode with spray voltage at 2.445 kV, and the ion transfer tube temperature at 250 °C. The MS was operated in DIA mode. A scan cycle comprised a full MS scan (m/z range from 350 to 1650), with RF lens at 40%, AGC target set to custom, normalised AGC target at 300%, maximum injection time mode set to custom, maximum injection time at 20 ms, microscan set to 1 and source fragmentation disabled. MS survey scan was followed by MS/MS DIA scan events using the following parameters: multiplex ions set to false, collision energy mode set to stepped, collision energy type set to normalised, HCD collision energies set to 25.5, 27, and 30%, orbitrap resolution 30,000, first mass 200, RF lens 40%, AGC target set to custom, normalised AGC target 3000%, microscan set to 1 and maximum injection time 55 ms. Data for both MS scan and MS/MS DIA scan events were acquired in profile mode.

## Naive vs LPS +IL-4 (Normal/CF +/-Fluvastatin)

Peptides were analysed using single-shot Data Independent Acquisition (DIA). For each sample, 200 ng of peptide was injected onto a C18 reverse-phase chromatography system (Vanquish, Thermo Fisher Scientific) and electrosprayed into an Astral Orbitrap Mass Spectrometer (Thermo Fisher). The following buffers were used for liquid chromatography: buffer A (0.1% formic acid in Milli-Q water [v/v]) and buffer B (80% acetonitrile and 0.1% formic acid in Milli-Q water (v/v)). Samples were loaded onto a trap column (Pep Map Neo C18, 5 µm, 300 µm x 5 mm, Thermo Fisher Scientific) equilibrated in buffer A. The column (Pep MAP RSLC C18, 2 µm, 150 µm x 15 cm) was equilibrated with 4% buffer and peptides were eluted at a variable flow rate, starting at 1.3 µl/min to 0.8 µl/min from 4% buffer B to 8% buffer B in 0.1 min, then from 8% buffer B to 22.5% buffer B in 13 min, and from 22.5% b to 35% B in 6.90 min. The low rate is increased from 0.8 µl/min to 2 µl/min in 0.4 min with buffer B increasing from 35% to 55%. The column is finally washed at a flow of 2 µl/min with 99% buffer B. The column was kept at a constant temperature of 50 °C. The data was acquired using an easy spray source operated in positive mode with spray voltage at 2.0 kV, and the ion transfer tube temperature at 280 °C. The MS was operated in DIA mode. A scan cycle comprised a full MS scan (m/z range from 380 to 980), with orbitrap resolution at 2,400,000, RF lens at 40%, AGC target set to custom, normalised AGC target at 500%, absolute AGC value set to 5.00e6, maximum injection time 5 ms and microscan set to 1. MS survey scan was followed by MS/MS DIA scan events using the following parameters: DIA window type set to auto, isolation window 2, window overlap set to 0, window placement optimisation on, number of scan events 299, collision energy type normalised, HCD collision energy 25%, detector type Astral, scan range 150–2000, normalised AGC target 500%, absolute AGC target 5.000e4, maximum injection time 5 ms, microscan set to 1, loop control time, time 0.6 s. Data for both MS scan and MS/MS DIA scan events were acquired in profile mode.

## Proteomic data analysis

For comparing naive and stimulated B cells, raw mass spec data files were searched using Spectronaut version 19 (Biognosys). Data was analysed by DirectDIA. Raw mass spec data files were searched against a mouse database (Swissprot Trembl November 2023) with the following parameters: directDIA, false discovery rate set to 1%, protein N-terminal acetylation and methionine oxidation were set as variable modifications and carbamidomethylation of cysteine residues was selected as a fixed modification.

The following Spectronaut settings were used (*Baker et al., 2024*): all identification settings were set to 0.01 (precursor q-value cut-off, precursor PEP cut-off, protein FDR strategy [accurate], protein q-value cut-off [experiment], protein q-value cut-off [run], and protein PEP cut-off). The following quantification settings were used: 'Quant 2.0', the MS-Level quantity was set to 'MS2', imputation was disabled, major group Top N and minor group Top N were set as 'False' and cross run normalisation was set as 'False'.

Intensity values from Spectronaut were further analysed in Perseus (*Tyanova et al., 2016*) and the histone ruler method *Wiśniewski et al., 2014* used to estimate copy number and concentrations of the identified proteins. Fold change was calculated relative to levels in naive B cells and significance determined using unpaired Student's t tests on log10-transformed data (*Supplementary file 3* and

*Supplementary file 4*). The False Discovery Rate approach using the two-stage step up method, Benjamini, Krieger, and Yekutieli (*Benjamini et al., 2006*) with a Q value of 0.05 (*Supplementary file 3* and *Supplementary file 4*). Enrichment against the Gene Ontology (GO) Biological Processes, Uniprot Keyword Biological Process and *Kyoto Encyclopaedia of Genes and Genomes* (KEGG) databases. Biological Process GO terms (*Aleksander et al., 2023*) and KEGG pathways (*Kanehisa et al., 2016*) was carried out using the DAVID gene enrichment site (*Sherman et al., 2022*; *Supplementary file 1*).

## Statistical analysis

Unless otherwise stated, graphs show mean values + SD with individual replicates shown by symbols. Unpaired two-tailed Student's *t* test, one-way ANOVA, two-way ANOVA with post-hoc testing were performed in Prism, and the tests used are indicated in the figure legends. Full results of the ANOVA calculations are given in *Supplementary file 5*.

## Resource availability

### Lead contact

Further information and requests for resources should be directed to and will be fulfilled by the lead contact, Simon Arthur (j.s.c.arthur@dundee.ac.uk).

### Materials availability

This study did not generate any new unique reagents.

## Acknowledgements

The authors thank A Rennie and R Clarke from the Flow Cytometry Facility for cell sorting and advice on flow cytometry. We also thank the FingerPrints Proteomics facility for running our samples for mass spectrometry and for advice on proteomics. This research was supported by a Wellcome Trust (DMSC, UNS148487) and MRC (MR/N013735/1, MR) PhD studentship awards.

## Additional information

### Funding

| Funder | Grant reference number | Author |
| --- | --- | --- |
| Wellcome Trust | UNS148487 | Dana MS Cheung |
| Medical Research Council | MR/N013735/1 | Momchil Razsolkov |

The funders had no role in study design, data collection and interpretation, or the decision to submit the work for publication. For the purpose of Open Access, the authors have applied a CC BY public copyright license to any Author Accepted Manuscript version arising from this submission.

### Author contributions

Dana MS Cheung, Formal analysis, Funding acquisition, Validation, Investigation, Visualization, Writing – original draft, Writing – review and editing; Momchil Razsolkov, Formal analysis, Funding acquisition, Validation, Investigation; Fabrizia Bonacina, Investigation, Writing – review and editing; Stephen Andrews, Investigation; Megan C Sumoreeah, Methodology; Linda V Sinclair, Resources, Methodology, Writing – review and editing; Andrew JM Howden, Conceptualization, Supervision, Project administration, Writing – review and editing; J Simon C Arthur, Conceptualization, Supervision, Funding acquisition, Writing – original draft, Project administration, Writing – review and editing

### Author ORCIDs

Dana MS Cheung ⓘ https://orcid.org/0000-0003-4920-8876
Fabrizia Bonacina ⓘ https://orcid.org/0000-0002-2611-1177
Megan C Sumoreeah ⓘ https://orcid.org/0000-0002-3275-3344
Linda V Sinclair ⓘ https://orcid.org/0000-0003-1248-7189

Andrew JM Howden 🆔 https://orcid.org/0000-0002-4332-9469
J Simon C Arthur 🆔 https://orcid.org/0000-0002-8135-1958

### Ethics

Wildtype C57BL/6/J mice (Charles Rivers) aged between 8 and 12 weeks were used for the majority of experiments. Conditional SLC7A5 knockout mice bred to Vav-iCre mice Slc7a5$^{fl/fl}$/Tg(Vav1-iCre)$^{+/-}$ have been described previously and were maintained on a C57Bl6/J background (de Boer et al, 2003; Poncet et al, 2014; Sinclair et al, 2013). MyD88$^{-/-}$ (MyD88 KO) mice have been described previously and were maintained on a C57BL6/J background (Adachi et al, 1998). The mice were kept in individually ventilated cages under specific pathogen-free conditions. Mice were given free access to food (RM3 irradiated pelleted diet, Special Diet Services) and water. Animal rooms were maintained at 21°C and a humidity of 45 to 55% with a 12/12 h light cycle. Mice were sacrificed using a rising concentration of $CO_2$, and death was confirmed by either cervical dislocation or exsanguination. This work was performed under a U.K. Home Office Project Licence (70/7901) in compliance with UK Home Office Animals (Scientific Procedures) Act 1986 guidelines, and approved by the University of Dundee's Welfare and Ethical Use of Animals Committee.

Reviewer #1 (Public review): https://doi.org/10.7554/eLife.109093.3.sa1
Reviewer #2 (Public review): https://doi.org/10.7554/eLife.109093.3.sa2
Author response https://doi.org/10.7554/eLife.109093.3.sa3

## Additional files

### Supplementary files

Supplementary file 1. Excel file containing terms for protein enrichment too large to fit in a PDF. Related to *Figure 1H*, *Figure 6—figure supplement 1*.

Supplementary file 2. Serum tests related to *Figure 5*.

Supplementary file 3. Excel file containing proteomic data set (Naïve vs LPS +IL-4).

Supplementary file 4. Excel file containing proteomic data set (Naïve vs LPS +IL-4 (Normal/CF +/-Fluvastatin)).

Supplementary file 5. Statistics related to all figures.

MDAR checklist

### Data availability

All data generated or analysed during this study are included in the manuscript and supporting files. The mass spectrometry proteomics data (Naïve vs LPS + IL-4) has been deposited to the ProteomeXchange Consortium (http://proteomecentral.proteomexchange.org) via the PRIDE partner repository (*Perez-Riverol et al., 2025*) with the dataset identifier PXD057739. The mass spectrometry proteomics data (Naïve vs LPS + IL-4 (Normal/CF) +/- Fluvastatin) has been deposited to PRIDE with the dataset identifier PXD074568. Raw FACS and western blot files have been provided as source data.

The following datasets were generated:

| Author(s) | Year | Dataset title | Dataset URL | Database and Identifier |
| --- | --- | --- | --- | --- |
| Cheung DMS, Razsolkov M, Bonacina F, Andrews S, Sumoreeah M, Sinclair LV, Arthur SC | 2026 | Lipopolysaccharide stimulates dynamic changes in B cell metabolism to promote proliferation | https://www.ebi.ac.uk/pride/archive/projects/PXD057739 | PRIDE, PXD057739 |
| Cheung DMS, Razsolkov M, Bonacina F, Andrews S, Sumoreeah M, Sinclair LV, Arthur SC | 2026 | Lipopolysaccharide stimulates dynamic changes in B cell metabolism to promote proliferation | https://www.ebi.ac.uk/pride/archive/projects/PXD074568 | PRIDE, PXD074568 |

The following previously published datasets were used:

| Author(s) | Year | Dataset title | Dataset URL | Database and Identifier |
|---|---|---|---|---|
| James O, Sinclair LV, Brock AAC, Salerno F, Brenes A, Khameneh HJ, Pecoraro M, Guarda G, Howden AJM, Lefter N | 2024 | A proteomic map of B cell activation and its shaping by mTORC1, MYC and iron | https://proteomecentral.proteomexchange.org/cgi/GetDataset?ID=PXD070254 | ProteomeXchange, PXD070254 |
| Mostafavi S, Yoshida H, Moodley D, LeBoité H | 2015 | ImmGen Cytokines: Interferons | https://www.ncbi.nlm.nih.gov/geo/query/acc.cgi?acc=GSE75306 | NCBI Gene Expression Omnibus, GSE75306 |
| Tesi A, de Pretis S, Furlan M, Filipuzzi M | 2019 | An early Myc-dependent transcriptional program underlies enhanced macromolecular biosynthesis and cell growth during B-cell activation | https://www.ncbi.nlm.nih.gov/geo/query/acc.cgi?acc=GSE126340 | NCBI Gene Expression Omnibus, GSE126340 |

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

# Appendix 1

**Appendix 1—key resources table**

| Reagent type (species) or resource | Designation | Source or reference | Identifiers | Additional information |
|---|---|---|---|---|
| Genetic reagent (*Mus musculus*) | C57BL/6J (WT) | Charles River UK | | |
| Genetic reagent (*Mus musculus*) | $Slc7a5^{fl/fl}/Tg(Vav1-iCre)^{+/-}$ | **Poncet et al., 2014** | | |
| Genetic reagent (*Mus musculus*) | MyD88$^{-/-}$ | **Adachi et al., 1998** | | |
| Biological sample, *Escherichia coli* O26:B6 | Lipopolysaccharide | Sigma Aldrich | Catalogue number: L2654 | Concentration: 20 µg/ml |
| Biological sample, *Streptomyces filipinensis* | Filipin | Merck | Catalogue number: F9765 | Concentration: 1 µg/µl |
| Antibody | Anti-mouse/human CD11b (Rat, monoclonal, M1/70) | BioLegend | Catalogue number: 101204 RRID:AB_312787 | Concentration: 10 µg/ml |
| Antibody | Anti-mouse CD11c (Armenian hamster, monoclonal, N418) | BioLegend | Catalogue number: 117304 RRID:AB_313773 | Concentration: 10 µg/ml |
| Antibody | Anti-mouse CD3ε (Armenian hamster, monoclonal, 145–2 C11) | BioLegend | Catalogue number: 100304 RRID:AB_312669 | Concentration: 10 µg/ml |
| Antibody | Anti-mouse TER119 (Rat, monoclonal, TER-119) | BioLegend | Catalogue number: 116204 RRID:AB_313705 | Concentration: 10 µg/ml |
| Antibody | Anti-mouse CD19 (Rat, monoclonal, 6D5) | BioLegend | Catalogue number: 115512 RRID:AB_313647 | Fluorophore: APC Cell surface stain dilution: (1:300) |
| Antibody | Anti-mouse CD19 (Rat, monoclonal, 1D3) | BD Biosciences | Catalogue number: 557399 RRID:AB_396682 | Fluorophore: PE Cell surface stain dilution: (1:200) |
| Antibody | Anti-mouse CD19 (Rat, monoclonal, 1D3) | BioLegend | Catalogue number: 152403 RRID:AB_2629812 | Fluorophore: FITC Cell surface stain dilution: (1:200) |
| Antibody | Anti-mouse CD93 (Rat, monoclonal, AA4.1) | BioLegend | Catalogue number: 136510 RRID:AB_2275868 | Fluorophore: APC Cell surface stain dilution: (1:200) |
| Antibody | Anti-mouse CD98 (Rat, monoclonal, RL388) | BioLegend | Catalogue number: 128208 RRID:AB_2190813 | Fluorophore: PE Cell surface stain dilution: (1:200) |
| Antibody | Anti-mouse IgG1 (Rat, monoclonal, A85-1) | BD Biosciences | Catalogue number: 553443 RRID:AB_394862 | Fluorophore: FITC Cell surface stain dilution: (1:200) |
| Antibody | Phospho-Rb (Ser807/811) (Rabbit, monoclonal, D20B12) | Cell Signalling Technology | Catalogue number: 4277 RRID:AB_2797605 | Fluorophore: Alexa Fluor 488 Intracellular stain dilution: (1:300) |
| Antibody | Anti-rabbit IgG (H+L), F(ab')$_2$ Fragment (Goat, unknown clonality) | Cell Signalling Technology | Catalogue number: 4414 RRID:AB_10693544 | Fluorophore: Alexa Fluor 647 Conjugate Dilution: (1:1000) |

*Appendix 1 Continued on next page*

*Appendix 1 Continued*

| Reagent type (species) or resource | Designation | Source or reference | Identifiers | Additional information |
|---|---|---|---|---|
| Antibody | Phospho-STAT1 (Tyr701) (Rabbit, monoclonal, 58D6) | Cell Signalling Technology | Catalogue number: 9167 RRID:AB_561284 | Intracellular stain dilution: (1:1000) |
| Antibody | Phospho-p44/42 MAPK (Erk1/2) (Thr202/Tyr204) (Rabbit, polyclonal) | Cell Signalling Technology | Catalogue number: 9101 RRID:AB_331646 | Western blot dilution: (1:1000) |
| Antibody | p44/42 MAPK (Erk1/2) (Rabbit, polyclonal) | Cell Signalling Technology | Catalogue number: 9102 RRID:AB_330744 | Western blot dilution: (1:1000) |
| Antibody | Phospho-p38 MAPK (Thr180/Tyr182) (Rabbit, monoclonal, D3F9) | Cell Signalling Technology | Catalogue number: 4511 RRID:AB_2139682 | Western blot dilution: (1:1000) |
| Antibody | p38 MAPK (Rabbit, polyclonal) | Cell Signalling Technology | Catalogue number: 9212 RRID:AB_330713 | Western blot dilution: (1:1000) |
| Antibody | Phospho-Akt (Ser473) (Rabbit, monoclonal, D9E) | Cell Signalling Technology | Catalogue number: 4060 RRID:AB_2315049 | Western blot dilution: (1:1000) |
| Antibody | Phospho-p70 S6 Kinase (Thr389) (Rabbit, monoclonal, 108D2) | Cell Signalling Technology | Catalogue number: 9234 RRID:AB_2269803 | Western blot dilution: (1:1000) |
| Antibody | Phospho-S6 Ribosomal Protein (Ser235/236) (Rabbit, polyclonal) | Cell Signalling Technology | Catalogue number: 2211 RRID:AB_331679 | Western blot dilution: (1:1000) |
| Antibody | S6 Ribosomal Protein (Rabbit, monoclonal, 5G10) | Cell Signalling Technology | Catalogue number: 2217 RRID:AB_331355 | Western blot dilution: (1:1000) |
| Antibody | Anti-Mouse IgM μ chain (Goat, polyclonal) | Stratech | Catalogue number: 115-006-075 RRID:AB_2338474 | Concentration: 10 μg/ml |
| Peptide, recombinant protein | IL-4 | Peprotech | Catalogue number: 214–14 | Concentration: 10 ng/ml |
| Peptide, recombinant protein | Resiquimod/R848 | Invivogen | Catalogue number: tlrl-r848 | Concentration:1 μg/ml |
| Peptide, recombinant protein | *Bléry et al., 2006* | Invivogen | Catalogue number: tlrl-2006 | Concentration:1 μg/ml |
| Peptide, recombinant protein | CD40 ligand | R&D | Catalogue number: 8230 CL-050/CF | Concentration: 500 ng/ml |
| Chemical compound, drug | O-propargyl-puromycin | Jena Bioscience | Catalogue number: NU-931–5 | Concentration: 20 μM |
| Chemical compound, drug | Saponin | Sigma Aldrich | Catalogue number: SAE0073 | Concentration: 0.01% |
| Chemical compound, drug | Copper sulphate | Sigma Aldrich | Catalogue number: 209198 | Concentration: 1 mM |
| Chemical compound, drug | Sodium ascorbate | Sigma Aldrich | Catalogue number: A7631 | Concentration: 10 mM |
| Chemical compound, drug | BTTAA | Sigma Aldrich | Catalogue number: 906328 | Concentration: 1 mM |
| Chemical compound, drug | Aminoguanidine | Cayman Chemicals | Catalogue number: 81530 | Concentration: 10 mM |

*Appendix 1 Continued*

| Reagent type (species) or resource | Designation | Source or reference | Identifiers | Additional information |
|---|---|---|---|---|
| Chemical compound, drug | Alexafluor 647 azide | Invitrogen | Catalogue number: A10277 | Concentration: 5 µM |
| Chemical compound, drug | L-Kynurenine | Sigma Aldrich | Catalogue number: K8625 | Concentration: 200 µM |
| Chemical compound, drug | 2-Amino-2-norbornanecarboxylic acid (BCH) | Sigma Aldrich | Catalogue number: A7902-1G | Concentration: 10 mM |
| Chemical compound, drug | Mevalonic acid 5-phosphate lithium salt hydrate | Sigma Aldrich | Catalogue number: 79849 | Concentration: 1 mM-2mM |
| Chemical compound, drug | Geranylgeranyl pyrophosphate ammonium salt | Sigma Aldrich | Catalogue number: G6025 | Concentration: 10 µM |
| Chemical compound, drug | Fluvastatin sodium | Tocris | Catalogue number: 3309 | Concentration: 0.1–100 µM |
| Chemical compound, drug | Rosuvastatin calcium | Tocris | Catalogue number: 6343 | Concentration: 0.1–100 µM |
| Chemical compound, drug | NB-598 | MedChem Express | Catalogue number: HY-16343 | Concentration: 0.1–100 µM |
| Chemical compound, drug | FGTI-2734 | MedChem Express | Catalogue number: HY-128350 | Concentration: 10 µM |
| Chemical compound, drug | FTI-277 hydrochloride | MedChem Express | Catalogue number: HY-15872A | Concentration: 10–30 µM |
| Chemical compound, drug | GGTI-289 trifluoroacetate | MedChem Express | Catalogue number: HY-15871 | Concentration: 10–30 µM |
| Chemical compound, drug | PD18352 | Tocris | Catalogue number: 4237 | Concentration: 2 µM |
| Chemical compound, drug | VX745 | Tocris | Catalogue number: 3915 | Concentration: 1 µM |
| Chemical compound, drug | Rapamycin | Merck | Catalogue number: R8781 | Concentration: 20 nM |
| Commercial assay, kit | CellTrace Violet Cell Proliferation Kit | Invitrogen | Catalogue number: C34557 | Concentration: 2.5 µM |
| Commercial assay, kit | MitoTracker Red FM | Invitrogen | Catalogue number: M22425 | Concentration: 100 nM |
| Software, algorithm | FlowJo software | BD Biosciences, developed by Treestar | RRID:SCR_008520 | Version 10.10.0 and above |
| Software, algorithm | Perseus | https://www.maxquant.org/perseus, PMID:27348712 | RRID:SCR_015753 | Version 1.6.6.0 |
| Software, algorithm | Spectronaut | Biognosys | | Version 19 |
| Software, algorithm | Prism | GraphPad | RRID:SCR_002798 | Version 9 or 10 |
| Other | Streptavidin Microbeads | Miltenyi Biotech | Catalogue number: 130-042-901 | Dilution: (1:10) |

