## [Editor Report · eLife Assessment]

Findings from this study are considered **fundamental** because they identify amino acid uptake, cholesterol synthesis, and protein prenylation as key metabolic regulators of B cell activation, proliferation, and survival, advancing understanding of T-independent immune responses. The study links metabolic reprogramming directly to B cell function, highlighting how cellular metabolism supports immune fitness. The evidence is **compelling**, combining unbiased proteomic profiling with genetic and pharmacological validation to demonstrate causal roles for these pathways.

---

## [Referee Report · Reviewer #1 (Public review)]

The work presented by Cheung et al. used a quantitative proteomics method to capture molecular changes in B cells exposed to LPS and IL-4, a combination of stimuli activating naive B cells. Amino acid transporters, cholesterol biosynthetic enzymes, ribosomal components, and other proteins involved in cell proliferation were found to increase in stimulated B cells. Experiments involving genetic loss-of-function (SLC7A5), pharmacological inhibition (HMGCR, SQLE, prenylation), and functional rescue by metabolites (mevalonate, GGPP) validated the proteomics data and revealed that amino acid uptake, cholesterol/mevalonate biosynthesis, and cholesterol uptake played a crucial role in B cell proliferation, survival, biogenesis, and immunoglobulin class switching. Experiments involving cholesterol-free medium showed that both biosynthesis and LDLR-mediated uptake catered to the cholesterol demand of LPS/IL-4-stimulated B cells. A role for protein prenylation in LDLR-mediated cholesterol uptake was postulated and backed by divergent effects of GGPP rescue in the presence and absence of cholesterol in culture medium.

Strengths:

The discovery was made by proteome-wide profiling and unbiased computational analysis. The discovered proteins were functionally validated using appropriate tools and approaches. The metabolic processes identified and prioritized from this comprehensive survey and systematic validation highly likely represent mechanisms of high importance and influence. Analysis of immune cell metabolism at the protein level is relatively compared to transcriptomic and metabolomic analysis.

The conclusions from functional validation experiments were supported by clear data and based on rational interpretations. This was enabled by well-established readouts/analytical methods used to determine cell proliferation, viability, size, cholesterol content, and transporter/enzyme function. The data generated from these experiments strongly support the conclusions.

This work reveals a complex, yet intriguing, relationship between cholesterol metabolism and protein prenylation as they serve to promote B cell activation. The effects of pharmacological inhibition and metabolite replenishment on the cholesterol content and activation of B cells were determined and logically interpreted.

Weaknesses:

The findings of this study were obtained almost exclusively from ex vivo B cell stimulation experiments. Their contribution to B cell state and B cell-mediated immune responses in vivo was not explored. Without in vivo data, the study still provides valuable mechanistic information and insights, but it remains unknown, and there is no discussion about, how the identified mechanisms may play out in B cell immunity.

The role of HMGCR, SQLE, and prenylation in B cell activation was assessed using pharmacological inhibitors. Evidence from other loss-of-function approaches, which could strengthen the conclusions, does not exist. This is a moderate weakness and somewhat offset by other data, including those obtained from the tests involving multiple distinct pharmacological inhibitors and the metabolite replenishment experiments.

---

## [Referee Report · Reviewer #2 (Public review)]

This study uses mass spectrometry to quantify how LPS + IL-4 modify the mouse B cell proteome as naïve cells undergo blastogenesis and enter the cell cycle. This analysis revealed changes in key proteins involved in amino acid transport and cholesterol biosynthesis. Genetic and pharmacological experiments indicated important roles for these metabolic processes in B cell proliferation.

This work provides new information about the regulation of TI B cell responses by changes in cell metabolism and also a comprehensive mass spectrometry dataset which will be an important general resource for future studies. The experiments are thorough and carefully carried out. The majority of conclusions are backed up by data that is shown to be highly significant statistically. The comprehensive mass spectrometry dataset will be an important general resource for future studies.

After revision, the study now includes new data showing that the up regulation of amino acid uptake and cholesterol metabolism is not restricted to LPS + IL-4 (TLR4 + IL4R) stimulation but is also observed after stimulation of TLR7, TLR9, CD40 and the BCR. This increases the impact of this work and shows that this metabolic rewiring is a common feature of B cell activation. The inclusion of inhibitor data showing important roles for mTOR and ERK/p38a MAP kinases in the metabolic changes identified and provides preliminary insights into the mechanisms involved.

---

## [Author Response]

The following is the authors’ response to the original reviews.

**Reviewer #1 (Public review):**

We agree with the reviewer that a limitation of our study is its focus on cell-based assays rather than in vivo experiments. We did consider evaluating the effects of statins on B cell responses in vivo; however, this approach is complicated by findings that statins can influence antigen presentation by dendritic cells, thereby impacting antibody responses (Xia et al, 2018). We have revised the discussion section to acknowledge this points.

The reviewer also noted that our study assessed the roles of HMGCR, SQLE, and prenylation in B cell activation using pharmacological inhibitors and genetic knockdown/out approaches. Loss-of-function techniques such as RNAi, siRNA, and CRISPR can be challenging to apply to primary B cells, but we are exploring their feasibility for future revisions. While we acknowledge the limitations of using pharmacological inhibitors, we have taken several steps to mitigate these, including targeting multiple steps in the cholesterol biosynthetic pathway using structurally distinct inhibitors and conducting rescue experiments by supplementing downstream metabolites. To strengthen the results on prenylation further, we have added data using two further distinct prenylation inhibitors (revised Figure 6I-P). To further investigate potential off-target effects of statins, we performed proteomic analysis of B cells treated with and without fluvastatin. The data suggest that fluvastatin primarily affects cholesterol metabolism and does not cause widespread off-target effects (new Figure 6 - figure supplement 1).

**Reviewer #1 (Recommendations for the authors):**
What signalling mechanisms link LPS sensing to proteomic and metabolic changes? Do these changes depend on specific signalling modules downstream of TLR4 (e.g., MyD88, TRIF, NF-kappaB, MAPKs)? Other receptors found to produce similar effects (TLR7, TLR9, CD40) may share these modules. This information could strengthen the conclusion by showing the chain of molecular events through which immune stimuli reprogram B cell metabolism.

Signalling through most TLRs, including TLR4, TLR7 and TLR9, requires the adaptor protein MyD88. To determine if MyD88 is required for LPS-induced signalling, we carried out immunoblotting to compare signalling in B cells between WT mice and MyD88-deficient mice. We found that phosphorylation of key downstream proteins, including p38 and ERK1/2 (MAPK signalling), Akt, p70S6K and S6 (mTOR signalling) was diminished in MyD88-deficient mice. These results have been added to the manuscript as Figure 9 - Figure supplement 1.

We assessed the requirement of these signalling pathways for LPS-induced proliferation by treating B cells with rapamycin to block mTORC1, PD184352 for MEK1/MEK2 (the upstream activators of ERK1/2), VX745 for p38 or a combination of PD184352 and VX745. These results have been added to the manuscript as the new Figure 9. Rapamycin demonstrated the strongest inhibitory effect on proliferation, and combinatorial blocking of MAPK signalling mildly reduced proliferation (Figure 9A-B). In terms of cholesterol metabolism, treatment with all of these inhibitors reduced cholesterol levels; however, treatment with PD184352 and VX745 reduced cholesterol to the same level as naïve B cells (Figure 9F).

Other activating stimuli appear to have similar effects, we showed originally that TLR7 and TLR9 activation had a similar effect on proliferation and cholesterol to TLR4, as did activation of CD40 and the BCR (Figure 10). We have now expanded this and shown that these other receptors can also promote protein synthesis (new Figure 3 - Figure supplement 1).

There seem to be errors in the manuscript text.(1) Page 6, line 232: ssRNAseq?

We appreciate the reviewer for spotting these issues. This has been amended to scRNAseq.

(2) Page 13, line 490: SC7A5?

This has been amended to SLC7A5

(3) The abbreviation CF (cholesterol-free?) is not defined when it first appears.

This has been amended to cholesterol-free (CF) on page 9, line 411.

**Reviewer #2 (Public review):**

The reviewer suggested that the study would be strengthened by determining whether the observed changes are specific to LPS + IL-4 stimulation or represent a more general B cell response to mitogenic signals. We believe that these effects are not specific to LPS and also occur with other mitogenic stimuli. We have expanded on the data in the original draft showing that other TLR agonists as well as CD40 and BCR stimulation increase both B cell proliferation and cholesterol levels and also looked at the effects of these stimuli on protein synthesis.

**Reviewer #2 (Recommendations for the authors):**
(1) One of the most highly enriched processes is 'response to interferon alpha'. This stands out as most of the other processes identified involve more general cellular processes (i.e., cell proliferation, cell metabolism, etc...). Minimally, interferon alpha should be discussed. It would also be interesting to test whether type I interferons regulate any of the metabolic changes identified.

Response to interferon alpha has the highest fold enrichment of 6.78. To look at this further compiled a list of proteins upregulated by IFN-α stimulation in murine B cells, derived from (Mostafavi et al, 2016) and compared these with our proteome. We found that most of the IFNα regulated genes were not significantly upregulated following LPS + IL-4 stimulation compared to naïve B cells (Figure 1 - Figure supplement 3A). We also measured phosphorylation of the transcription factor STAT1, which is induced by IFNα and IFNβ signalling, and found that LPS stimulation did not induce p-STAT1 (Figure 1 - Figure supplement 3B-C). These results have been added to the manuscript as Figure 1 - Figure supplement 3. Despite this, as discussed further in the manuscript we cannot rule out a weak interferon response in the proteomics.

(2) The proteome of BCR-stimulated B cells has been analyzed by mass spectrometry. This dataset should be compared with the LPS + IL-4 dataset of the current study. This may reveal whether these two stimulations have similar or different effects on B-cell function. In particular, it is interesting to know whether BCR stimulation induces SLC7A5 expression and whether proteins involved in cholesterol metabolism are altered by BCR stimulation.

A similar study using anti-IgM and anti-CD40 to activate murine B cells has found an upregulation of amino acid transporters, including SLC7A5, in their proteomic data, suggesting that this is not a stimulus-specific effect. This has been added to the text subsection “Protein synthesis in LPS + IL-4 stimulated B cells is dependent on the uptake of amino acids.” In line with this we have also shown that stimulation of the BCR upregulates protein synthesis (new Figure 3 - Figure supplement 1). We have added data on HMGCR, SQLE and LDLR from the IL-4, anti-IgM and anti-CD40 a proteomic experiments to the new Figure 10 - Figure supplement 1. As the BCR proteome published as a preprint (James et al 2026) is about to be resubmitted as a distinct paper that does not deal with cholesterol metabolism, we have not expanded on this dataset further.

(3) A role for mTORC1 has been shown for proteome remodelling following BCR stimulation of naïve B cells, regulating the expression of amino acid transporters. Is mTORC1 involved in any of the changes detected following LPS + IL-4 stimulation? (i.e., cell proliferation, ribosome biogenesis, amino acid transport, cholesterol biogenesis).

To determine the importance of mTORC1 for B cell function, we treated B cells with rapamycin. We found that rapamycin treatment slightly reduced protein synthesis (Figure 9 - Figure supplement 2A) and amino acid uptake (Figure 9 - Figure supplement 2B). These results have been added to the manuscript as Figure 9 - Figure supplement 2. Rapamycin reduced cholesterol to almost the levels in naïve B cells (new Figure 9F) and had a significantly inhibitory effect on proliferation (new Figure 9A-B).

(4) Analysis of Slc7a5 knockout B cells showed that SLC7A5 is required for LPS-induced proliferation (Figure 4G). Is SLC7A5 required for B cell growth following LPS + IL-4 stimulation? Is SLC7A5 required for BCR-induced B cell proliferation/growth?

There appears to be a misunderstanding, as Figure 4G compares proliferation between WT and SLC7A5 KO B cells following LPS + IL-4 stimulation and not LPS stimulation alone.

Unfortunately, we no longer have access to *Slc7a5fl/fl/Vav-iCre+/-* mice and will not be able to measure CTV staining for proliferation following BCR stimulation. However, a similar study using anti-IgM and anti-CD40 to activate murine B cells found that B cells from *Slc7a5fl/fl/Vav-iCre+/-* mice were significantly smaller, had reduced expression of the chaperone protein CD98 and impaired expression of the transferrin receptor CD71, which is required for iron uptake, compared to WT B cells (James et al, 2026).

(5) The expression of several key proteins (regulating proliferation/amino acid transport/cholesterol metabolism) is shown to be significantly upregulated by LPS + IL-4 stimulation of naïve B cells. It would be interesting to determine whether these increases result from induced transcription of the relevant genes. This could initially be assessed by qRT-PCR analysis of LPS + IL-4 stimulated primary B cells, or alternatively, mining of online RNAseq datasets.

We mined RNA-Seq data from C57BL/6 mice (Tesi et al, 2019) which compared naïve B cells and B cells after 2,4, or 8 hours of LPS stimulation. We found that the transcription of genes that coded for the amino acid transporter SLC7A5/SLC3A2 (Figure 4 - Figure supplement 2A-B) and key genes involved in cholesterol metabolism followed the same pattern of upregulation as our proteomic data (Figure 4 - Figure supplement 2C-F). These results have been added to the manuscript as a new Figure 4 - Figure supplement 2.

(6) Cholesterol levels are shown to be increased following resiquimod, CpG, anti-IgM, and CD40L stimulation (Figure 9). What effect do these agonists have on levels of HMGCR, SQLE, and LDLR in B cells? Is B-cell growth by these agonists impaired by Fluvastatin.

We found that stimulation of murine B cells with either IL-4, anti-IgM or anti-CD40 could increase levels of HMGCR, SQLE and LDLR, with the largest increase seen with a combination of these stimuli (Figure 10 - Figure supplement 1A-D) (James et al, 2026). These results have been added to the manuscript as Figure 10 - Figure supplement 1.

Figures 10C-E show that B cell growth, survival and proliferation are impaired by Fluvastatin after Resiquimod, CpG, anti-IgM, and CD40L stimulation, although we do not have proteomic data from these stimuli to confirm the levels of HMGCR, SQLE and LDLR.

We carried out proteomics after 24 hours of LPS + IL-4 stimulation in normal/CF media, with or without Fluvastatin. We found that Fluvastatin treatment in normal media increased the expression of HMGCR, SQLE and LDLR. Fluvastatin treatment in CF media had the highest increase in the expression of these key proteins (Figure 6 - Figure supplement 1G-I). These results have been added to the manuscript as Figure 6 - Figure supplement 1.

(7) Do Fluvastatin or FGTI-2734 affect early activation of signaling pathways by LPS + IL-4 stimulation of B cells? (eg. MAPKs, STATs, PI3K/AKT).

This is an interesting question, we will pursue this in our future work.

References:

James O, Sinclair LV, Lefter N, Brock AAC, Salerno F, Brenes A, Khameneh HJ, Pecoraro M, Guarda G & Howden AJM (2026) A proteomic map of B cell activation and its shaping by mTORC1, MYC and iron. 2024.12.19.629506 doi:10.1101/2024.12.19.629506 [PREPRINT]

Mostafavi S, Yoshida H, Moodley D, LeBoité H, Rothamel K, Raj T, Ye CJ, Chevrier N, Zhang S-Y, Feng T, *et al* (2016) Parsing the Interferon Transcriptional Network and Its Disease Associations. *Cell* 164: 564–578

Tesi A, de Pretis S, Furlan M, Filipuzzi M, Morelli MJ, Andronache A, Doni M, Verrecchia A, Pelizzola M, Amati B, *et al* (2019) An early Myc‐dependent transcriptional program orchestrates cell growth during B‐cell activation. *EMBO reports* 20: e47987

Xia Y, Xie Y, Yu Z, Xiao H, Jiang G, Zhou X, Yang Y, Li X, Zhao M, Li L, et al (2018) The Mevalonate Pathway Is a Druggable Target for Vaccine Adjuvant Discovery. Cell 175: 1059-1073.e21